# Nonasymptotic Guarantees for Spiked Matrix Recovery with Generative Priors

**Jorio Cocola**
Department of Mathematics
Northeastern University
Boston, MA 02115
cocola.j@northeastern.edu

**Paul Hand**
Department of Mathematics
and Khoury College of Computer Sciences,
Northeastern University
Boston, MA 02115
p.hand@northeastern.edu

**Vladislav Voroninski**
Helm.ai,
Menlo Park, CA 94025
vlad@helm.ai

## Abstract

Many problems in statistics and machine learning require the reconstruction of a rank-one signal matrix from noisy data. Enforcing additional prior information on the rank-one component is often key to guaranteeing good recovery performance. One such prior on the low-rank component is sparsity, giving rise to the sparse principal component analysis problem. Unfortunately, there is strong evidence that this problem suffers from a computational-to-statistical gap, which may be fundamental. In this work, we study an alternative prior where the low-rank component is in the range of a trained generative network. We provide a non-asymptotic analysis with optimal sample complexity, up to logarithmic factors, for rank-one matrix recovery under an expansive-Gaussian network prior. Specifically, we establish a favorable global optimization landscape for a nonlinear least squares objective, provided the number of samples is on the order of the dimensionality of the input to the generative model. This result suggests that generative priors have no computational-to-statistical gap for structured rank-one matrix recovery in the finite data, nonasymptotic regime. We present this analysis in the case of both the Wishart and Wigner spiked matrix models.

## 1 Introduction

In this paper we study the problem of estimating a spike vector $y_\star \in \mathbb{R}^n$ from data $Y$ consisting of a rank-1 matrix perturbed with random noise. In particular, the following random models for $Y$ will be considered.

- The **Spiked Wishart Model** in which $Y \in \mathbb{R}^{N \times n}$ is given by:
$$Y = u\, y_\star^\mathsf{T} + \sigma \mathcal{Z}, \tag{1}$$
where $\sigma > 0$, $u \sim \mathcal{N}(0, I_n)$ and $\mathcal{Z}$ are independent and $\mathcal{Z}_{ij}$ are i.i.d. from $\mathcal{N}(0,1)$.

- The **Spiked Wigner Model** in which $Y \in \mathbb{R}^{n \times n}$ is given by:
$$Y = y_\star y_\star^\mathsf{T} + \nu \mathcal{H} \tag{2}$$
where $\nu > 0$, $\mathcal{H} \in \mathbb{R}^{n \times n}$ is drawn from a *Gaussian Orthogonal Ensemble* GOE$(n)$, i.e. $\mathcal{H}_{ii} \sim \mathcal{N}(0, 2/n)$ for all $1 \le i \le n$ and $\mathcal{H}_{ij} = \mathcal{H}_{ji} \sim \mathcal{N}(0, 1/n)$ for $1 \le j < i \le n$.

Spiked random matrices have been extensively studied in recent years as they serve as a mathematical model for many statistical inverse problems such as PCA [34, 2, 21, 58], synchronization over graphs [1, 8, 33] and community detection [43, 20, 47]. They are, moreover, connected to the rank-1 case of other linear inverse problems such as matrix sensing and matrix completion under RIP-like assumptions on the measurements operator [12, 65].

In the high-dimensional/low signal-to-noise ratio regimes, it is fundamental to leverage additional prior information on the low-rank component in order to obtain consistent estimates of $y_\star$. Recent works, however, have discovered that some priors give rise to gaps between what is statistically-theoretically optimal and can be achieved with unbounded computational resources, and what instead can be achieved with polynomial-time algorithms. A prominent example is represented by the Sparse PCA problem in which the vector $y_\star$ in (1) is taken to be sparse (see next section and [9, 37] for surveys of recent approaches).

In this paper we study the spiked random matrix models (1) and (2), where the prior information on the planted signal $y_\star$ comes from a learned generative network. In particular, we assume that a generative neural network $G : \mathbb{R}^k \to \mathbb{R}^n$ with $k < n$, has been trained on a data set of spikes, and the unknown spike $y_\star \in \mathbb{R}^n$ lies on the range of $G$, i.e. we can write $y_\star = G(x_\star)$ for some $x_\star \in \mathbb{R}^k$. As a mathematical model for the trained $G$, we consider a $d$-layer feed forward network of the form:

$$G(x) = \mathrm{relu}(W_d \ldots \mathrm{relu}(W_2 \mathrm{relu}(W_1 x)) \ldots) \qquad (3)$$

with weight matrices $W_i \in \mathbb{R}^{n_i \times n_{i-1}}$ and $\mathrm{relu}(x) = \max(x, 0)$ is applied entrywise. We furthermore assume that the network is expansive, i.e. $n = n_d > n_{d-1} > \cdots > n_0 = k$, and the weights have Gaussian entries. This modeling assumption was introduced in [29], and additionally it and its variants were used in [31, 26, 41, 25, 57]. See Section 1.1 for justifications of this model.

Generative priors have been shown to close a computational-to-statistical gap in the Compressive Phase Retrieval problem. With a sparsity prior the information-theoretically optimal sample complexity is proportional to the sparsity level $s$ of the signal, on the other hand the best known algorithms (convex methods [28, 39, 48], iterative thresholding [15, 60, 64], etc.) require a sample complexity proportional to $s^2$ for stable recovery, a barrier which might not be resolvable by polynomial-time algorithms [10]. Under the generative prior (3), [26] has shown that, compressive phase retrieval is possible via gradient descent over a nonlinear objective with sample complexity proportional (up to log factors) to the underlying signal dimensionality $k$. This result suggests that it may be possible to use generative priors to close other computational-to-statistical gaps such as for models (1) and (2). Indeed, recently [7] considered these low-rank models and the generative network prior (3) and shows that in the asymptotic limit $k, n, N \to \infty$ with $n/k = \mathcal{O}(1)$ and $N/n = \mathcal{O}(1)$, an Approximate-Message Passing algorithm achieves the statistical information-theoretic lower bound and no computational-to-statistical gap is present.

This paper analyzes the low-rank matrix models (1) and (2) under the generative network prior (3).

The contributions of this paper are as follows. We analyze the global landscape of a natural least-square loss over the range of the generative network demonstrating its benign optimization geometry. Our result provide further evidences for the claim that rank-one matrix recovery does not have computational-to-statistical gaps when enforcing a generative prior in the non-asymptotic finite-data regime. This provides a second problem for which generative priors have closed such gaps in a non-asymptotic case. We further corroborate these findings by proposing a (sub)gradient algorithm which, as shown by our numerical experiments, is able to recover the sought spike with optimal sample complexity. This paper, therefore, strengthens the case for generative networks as priors for statistical inverse problems, not only because of their ability to learn natural signal priors, but also because of their capacity to lead to statistically optimal polynomial-time algorithms and zero computational-to-statistical gaps.

## 1.1  Problem formulation and main results

We consider the rank-one matrix recovery problem under a deep generative prior. We assume that the signal spike lies in the range of the generative prior $y_\star = G(x_\star)$. To estimate $y_\star$, we propose to first find an estimate $\hat{x}$ of the latent variable $x_\star$ and then use $G(\hat{x}) \approx y_\star$. We thus consider the following

minimization problem[1]:

$$\min_{x \in \mathbb{R}^k} f(x) := \frac{1}{4}\|G(x)G(x)^\mathsf{T} - M\|_F^2. \tag{4}$$

where:

- for the **Wishart model** (1) we take $M = \Sigma_N - \sigma^2 I_n$ with $\Sigma_N = Y^\mathsf{T} Y / N$.
- for the **Wigner model** (2) we take $M = Y$.

Despite the objective function (4) being nonconvex and nonsmooth, we show that it enjoys a favorable global optimization geometry for Gaussian weight matrices $\{W_i\}_{i=1}^d$. The informal version of our main results for the two spiked models is given below.

**Theorem 1** (Informal). *Let $y_\star = G(x_\star)$ for a given a generative network $G : \mathbb{R}^k \to \mathbb{R}^n$ as in (3). Assume that each layer is sufficiently expansive, i.e. $n_{i+1} = \Omega(n_i \log n_i)$, and the weights are Gaussian. Consider the minimization problem* (4) *and assume that up to factors dependent on the number of layers $d$:*

- *for the **Wishart model**: $\sqrt{k \log n / N} \lesssim 1$,*

- *for the **Wigner model**: $\nu\sqrt{k \log n / n} \lesssim 1$.*

*With high probability:*

A. *for any nonzero point $x \in \mathbb{R}^k$ outside two small neighborhoods of $x_\star$ and $-\rho_d x_\star$ with $0 < \rho_d \le 1$, the objective function* (4) *has a direction of strict descent given almost everywhere by the gradient of $f$;*

B. *the objective function values near $-\rho_d x_\star$ are larger than those near $x_\star$, while $x = 0$ is a local maximum;*

C. *for any point $x$ in the small neighborhood around of $x_\star$, up to polynomials in $d$:*

- *for the **Wishart model**:*

$$\|G(x) - y_\star\|_2 \lesssim \sqrt{\frac{k \log n}{N}}, \tag{5}$$

- *for the **Wigner model**:*

$$\|G(x) - y_\star\|_2 \lesssim \nu\sqrt{\frac{k \log n}{n}}. \tag{6}$$

Our main result characterizes the global optimization geometry of the problem (4) for a network with an expansive architecture and Gaussian weights. Even though the objective function in (4) is a piecewise-quartic polynomial, we show that outside two small neighborhoods around $x_\star$ and a negative multiple of it, there are no other spurious local minima or saddles, and every nonzero point has a strict linear descent direction. The point $x = 0$ is a local maximum and a neighborhood around $x_\star$ contains the global minimum of $f$.

We note, moreover, that for any point $x$ in the "benign neighborhood" of $x_\star$, the reconstruction error $\|G(x) - y_\star\|$ has information-theoretically optimal rates (5) and (6) corresponding (up to log factors) to the best achievable even in the simple case of a $k$-dimensional subspace prior. This implies that for the Wishart model the number of samples required to estimate $y_\star$ scales like the latent dimension $k$ which corresponds to the intrinsic degrees of freedom of the signal $y_\star$. Similarly for the Wigner model this implies that enforcing the generative network prior leads to a reduction of the noise by a factor of $k/n$.

Furthermore we observe that the direction of descent guaranteed by the theorem are almost everywhere given by the gradient of the objective function $f$. Our result, therefore, suggests that spiked matrix recovery with a deep (random) generative network prior can be solved rate-optimally by simply

minimizing over the range of the network via simple and computationally tractable algorithms such as gradient descent methods. For small enough step sizes, the iterates of these methods would converge to one of the two neighborhoods where the gradients are small (identified in Theorem 1A), and avoiding the bad neighborhood of $-\rho_d x_\star$ can be done by exploiting the knowledge of the properties of the loss function (described in Theorem 1B) as done in Algorithm 1 below and shown in the numerical experiments. Finally, proving a convexity-like property of the "benign neighborhood" around $x_\star$ would ensure that the iterates will remain in this neighborhood and gradient descent will converge to a point with optimal error-rates (5) and (6). Formally proving the optimality and polynomial runtime of a gradient method for spiked matrix recovery is left for future work.

Regarding the Gaussian weight assumption, we observe that there is empirical evidence that the distribution of the weights of deep neural networks have properties consistent with those of Gaussian matrices [4]. Moreover, these observations have been used in advancing the theoretical understanding of deep network trained in supervised setting and in particular their ability to preserve the metric structure of the data [24]. The randomness assumption has been further used by [3] to show that autoencoders with random weights can be learned in polynomial time. More recently, a series of works (see for example [40, 23, 51, 44, 17]) have been dedicated to theoretical guarantees for training deep neural networks in the close-to-random regime of the *Neural Tangent Kernel* [32]. Finally, as for the case of compressed sensing in which the analysis of the random setting has led to considerable understanding of the problem as well as tangible practical innovations, we hope that the analysis of the random setting for deep generative networks will provide insights and generate novel developments in the field of statistical inverse problems.

We finally observe that signal recovery problems where multiple signal structures hold simultaneously, e.g. low-rankness and sparsity, have been notoriously difficult, leading to no tractable algorithms at optimal sample complexity (see the next section for further details). Consequently, one might expect that enforcing low-rankness and generative priors would be comparably hard. In this work, we show instead that this combination of structural priors is not inherently difficult. This would motivate practitioners to invest in building and using generative priors, as those studied in this paper, in contexts where other priors have been traditionally used with suboptimal theoretical guarantees or empirical performance.

## 2 Related work

**Sparse PCA and other computational-to-statistical gaps.**

Given a large number of samples data $\{y_i\}_{i=1}^N \in \mathbb{R}^n$ the important statistical task of finding the directions that explain most of the variance (principal components) is classically solved by PCA. Insights on the statistical performance of this algorithm can be gained by studying spiked covariance models [34]. Under this model it is assumed that the data are of the form:

$$y_i = u_i y_\star + \sigma z_i \tag{7}$$

where $\sigma > 0$, $u_i \sim \mathcal{N}(0,1)$ and $z_i \sim \mathcal{N}(0, I_n)$ are independent and identically distributed, and $y_\star$ is the unit norm planted spike. Note that a matrix $Y \in \mathbb{R}^{N \times n}$ with rows $\{y_i\}_i$ can be written as (1), and the $y_i$s are i.i.d. samples of $\mathcal{N}(0, \Sigma)$ where the population covariance matrix is $\Sigma = y_\star y_\star^\mathsf{T} + \sigma^2 I_n$. Principal Component Analysis, then, estimates $y_\star$ via the leading eigenvector $\hat{y}$ of the empirical covariance matrix $\Sigma_N = \frac{1}{N} \sum_{i=1}^N y_i y_i^\mathsf{T}$. Standard techniques of high dimensional probability then show that as long as[2] $N \gtrsim n$, with overwhelming probability:

$$\min_{\epsilon = \pm} \| \epsilon \hat{y} - y_\star \|_2 \lesssim \sqrt{\frac{n}{N}}. \tag{8}$$

However, in the modern high dimensional data regime, it is not uncommon to consider cases where the ambient dimension of the data $n$ is larger, or of the order, of the number of samples $N$. In this case, bounds of the form (8) become meaningless. Even worse, in the asymptotic regime $n/N \to c > 0$ and for $\sigma^2$ large enough, the spike $y_\star$ and the estimate $\hat{y}$ become orthogonal [35]. Moreover, minimax

techniques can be used to show that in this regime no other estimators can achieve better overlap with $y_\star$ [59].

These negative results motivated the use of additional structural prior on the spike $y_\star$, aimed at reducing the sample complexity of the problem. In recent years various priors has been studied such as positivity [46], cone constraints [22] and in particular sparsity [35], [66]. In the latter case $y_\star$ is assumed to be $s$-sparse, and it can be shown that for $N \gtrsim s \log n$ and $n \gtrsim s$, then the $s$-sparse largest eigenvector $\hat{y}_s$ of $\Sigma_N$:

$$\hat{y}_s = \operatorname*{argmax}_{y \in \mathcal{S}_2^{n-1}, \|y\|_0 \leq s} y^\mathsf{T} \Sigma_N y$$

satisfies the condition:

$$\min_{\epsilon = \pm} \|\epsilon \hat{y}_s - y_\star\|_2 \lesssim \sqrt{\frac{s \log n}{N}}.$$

In particular the number of samples must scale linearly with the intrinsic dimension $s$ of the signal. These rates are also minimax optimal, see for example [58] for the mean squared error and [2] for the support recovery. Despite these encouraging results no known polynomial time algorithm is known that achieves such performances and for example the covariance thresholding algorithm of [36] requires $N \gtrsim s^2$ samples in order to obtain exact support recovery or estimation rate:

$$\min_{\epsilon = \pm} \|\epsilon \hat{y}_s - y_\star\|_2 \lesssim \sqrt{\frac{s^2 \log n}{N}},$$

as shown in [21]. In summary, only computationally intractable algorithms are known to reach the statistical limit $N = \Omega(s)$ for Sparse PCA, while polynomial time methods are only sub-optimal requiring $N = \Omega(s^2)$. The study of this computational-to-statistical gap was initiated by [11] who investigated the detection problem via a reduction to the planted clique problem which is conjectured to be computationally hard.

The hardness of sparse PCA has been further suggested in a series of recent works [16, 42, 38, 14]. These works fit in the growing and important body of literature on computational-to-statistical gaps, which have also been found and studied in a variety of other contexts such as tensor principal component analysis [54], community detection [19] and synchronization over groups [52]. Many of these problems can be phrased as recovery a spike vector from a spiked random matrix models, and the hardness can then be viewed as arising from imposing simultaneously low-rankness and additional prior information on the signal (sparsity in case of Sparse PCA). This difficulty can be found in sparse phase retrieval as well, where [39] has shown that for an $s$-sparse signal of dimension $n$ lifted to a rank-one matrix, $m = \mathcal{O}(s \log n)$ number of quadratic measurements are enough to ensure well-posedness, while $m \geq \mathcal{O}(s^2 / \log^2 n)$ measurements are necessary for the success of natural convex relaxations of the problem. Similarly [50] studies the recovery of simultaneously low-rank and sparse matrices, and show the existence of a gap between what can be achieved with convex and tractable relaxations and nonconvex and intractable methods.

**Recovery with a generative network prior**

Recently, in the wake of successes of deep learning , deep generative networks have gained popularity as a novel approach for encoding and enforcing priors. They have been successfully used as a prior for various statistical estimation problems such as compressed sensing [13], blind deconvolution [5], inpainting [63], and many more [56, 62, 53, 61], etc.

Parallel to these empirical successes, a recent line of works have investigated theoretical guarantees for various statistical estimation tasks with generative network priors. Following the work of [13], [30] have given global guarantees for compressed sensing, followed then by many others for various inverse problems [55, 45, 25, 6, 53]. In particular [27] have shown that $m = \Omega(k \log n)$ number of measurements are sufficient to recover a signal from random phaseless observations, assuming that the signal is the output of a trained generative network with latent space of dimension $k$. Note that, contrary to the sparse phase retrieval problem, generative priors for phase retrieval allow optimal sample complexity, up to logarithmic factors, with respect to the intrinsic dimension of the signal. Further, when modeled by generative priors, that dimensionality could be much smaller than the sparsity level $s$ under a sparsity prior and an appropriate basis.

Recently [7] has shown that when $y_\star$ is in the range of an expansive-Gaussian generative network with Relu activation functions, then low-rank matrix recovery does not have a computational-to-statistical

gap, in the asymptotic limit $k, n, N \to \infty$ with $n/k = \mathcal{O}(1)$ and $N/n = \mathcal{O}(1)$. They also provide a spectral algorithm and demonstrate that it is able to match asymptotically the information-theoretical optimal. These methods were then extended to the phase-retrieval problem in [6].

## 3 Low-rank matrix recovery under a generative network prior

We are now ready to formulate our main theoretical result for the spiked random matrix models. Its analysis will be based on the following assumptions on the weights of the network.

**Assumption 1.** *The generative network $G$ defined in* (3), *has weights $W_i \in \mathbb{R}^{n_i \times n_{i-1}}$ with i.i.d. entries from $\mathcal{N}(0, 1/n_i)$ and satisfying the expansivity condition with constant $\epsilon > 0$:*

$$n_{i+1} \geq c\epsilon^{-2} \log(1/\epsilon) n_i \log n_i$$

*for all $i$ and a universal constant $c > 0$.*

We remark that no assumption on the layer-wise independence of the weight matrices is required.

Due to the non-smoothness of the Relu activation functions, the generative network $G$ and the loss function $f$ are not differentiable everywhere. Therefore, following [31], we resort to some concepts from nonsmooth analysis [3]. Since $f$ is continuous and piecewise smooth, at every point $x \in \mathbb{R}^k$, $f$ has a *Clarke subdifferential* given by:

$$\partial f(x) = \text{conv}\{v_1, v_2, \ldots, v_T\} \tag{9}$$

where conv denotes the convex hull of the vectors $v_1, \ldots, v_T$, gradients of the $T$ smooth functions adjoint at $x$. In particular at a point where $f$ is differentiable $\partial f(x) = \{\nabla f(x)\}$. The terms subgradients will be used for the vectors $v_x \in \partial f(x)$.

The next theorem will demonstrate the favorable optimization geometry of the minimization problem (4) for the spiked matrix models (1) and (2). We will show that provided that the noise scales linearly with the latent dimension $k$, outside 0 and two small Euclidean balls around $x_\star$ and a negative multiple of $x_\star$, the subgradient $v_x$ give a descent direction for the function $f(x)$. We let $\mathcal{B}(x, r)$ denotes the Euclidean ball of radius $r$ around $x$, and $D_v f(x)$ denotes the (normalized) one-sided directional derivative of $f$ in direction $v$: $D_v f(x) = \lim_{t \to 0} \frac{f(x+tv) - f(x)}{t \|v\|_2}$.

**Theorem 2** (Global Landscape Analysis)**.** *Let Assumption 1 be satisfied with $\epsilon \leq K_1 d^{-96}$, consider the minimization problem* (4) *and assume that the noise variance $\omega$ satisfies $\omega \leq K_2 \|x_\star\|_2^2 2^{-d}/d^{44}$ where:*

- *for the **Spiked Wishart Model** (1) take $M = \Sigma_N - \sigma^2 I_n$ with $\Sigma_N = Y^\intercal Y/N$ and:*

$$\omega := (\|y_\star\|_2^2 + \sigma^2) \max \left\{ \sqrt{\frac{113k \log(3\, n_1^d n_2^{d-1} \ldots n_{d-1}^2 n)}{N}}, \frac{52k \log(3\, n_1^d n_2^{d-1} \ldots n_{d-1}^2 n)}{N} \right\};$$

- *for the **Spiked Wigner Model** (2) take $M = Y$ and:*

$$\omega := \nu \sqrt{\frac{30k \log(3\, n_1^d n_2^{d-1} \ldots n_{d-1}^2 n)}{n}}.$$

*Then for $\gamma_\epsilon > 0$ depending polynomially on $\epsilon$, with probability at least $1 - 2e^{-k \log n} - \sum_{i=1}^d 8n_i e^{-\gamma_\epsilon n_{i-1}}$ the following holds.*

*For all $x \in \mathbb{R}^k$:*

- *if $x \notin \mathcal{B}(x_\star, r_+) \cup \mathcal{B}(-\rho_d x_\star, r_-) \cup \{0\}$ and $v_x \in \partial f(x)$:*

$$D_{-v_x} f(x) < 0$$

  *where*

$$r_+ = K_3(d^{14}\epsilon^{1/2} + 2^d d^{10}\omega \|x_\star\|_2^{-2}) \|x_\star\|_2,$$

  *and*

$$r_- = K_4(d^{12}\epsilon^{1/4} + 2^{d/2} d^{10}\omega^{1/2} \|x_\star\|_2^{-1}) \|x_\star\|_2$$

- *if $x \in \mathcal{B}(0, \|x_\star\|_2/16\pi) \backslash \{0\}$ and $v_x \in \partial f(x)$ then*
$$\langle x, v_x \rangle < 0$$
  *while if $x = 0$ and $v \in \mathcal{S}^{k-1}$ then*
$$D_{-v} f(0) = 0$$

*Here $K_1, \ldots, K_4$ are universal constants and $0 < \rho_d \le 1$ depends only on the depth $d$ and converges to 1 as $d \to \infty$.*

According to the theorem the subgradients of $f$ are direction of strict descent for any nonzero point outside the two Euclidean balls around $x_\star$ and $-x_\star$. Furthermore, there are no other spurious critical points or non-escapable saddles apart from the local maximum $x = 0$.

For small enough $\epsilon > 0$, the size of the two neighborhoods around $x_\star$ and $-x_\star$ is a function of the control parameter $\omega$. This quantity is analogous to the effective SNR $\sigma^2 \sqrt{s \log(n)/N}$ which governs sharp transitions in Sparse PCA [2]. In particular, for the Wishart model (structured PCA problem), in the interesting regime $k \log(n) \lesssim N$ and at fixed depth $d$, the size of the ball around $x_\star$ shrinks at the optimal rate $\sqrt{k \log(n)/N}$, implying that a number of samples $N$ proportional to $k$ is sufficient for a consistent estimate. In the same fashion, for the Wigner model the size of the noise $\nu$ it is required to be inversely proportional to the optimal rate $\sqrt{k/n}$.

We note that the quantity $2^d$ in the hypothesis and conclusions of the theorem, is an artifact of the scaling of the network and it should not be taken as requiring exponentially small noise. Indeed under the assumptions on the weights specified above, these matrices have spectral norm approximately 1, while the application of the Relu function zeros out approximately half of the entries of its argument leading to an "effective" operator norm of approximately $1/2$. The other polynomial dependence on the depth $d$ are likely not optimal and optimizing the proof for superior dependence on $d$ would not drastically alter the fundamental theoretical advance. As we show in the numerical experiments the bounds are quite conservative and the actual dependence on the depth is much better in practice.

Having analyzed the behavior of the loss $f$ outside the two balls around $x$ and $-\rho_d x_\star$, the next proposition will describe the local properties of these two neighborhoods.

**Proposition 1.** *Let the assumptions of Theorem 2 be satisfied.*

*A. For any $x \in \mathcal{B}(x_\star, r_+)$ and $y \in \mathcal{B}(-\rho_d\, x_\star, r_-)$ it holds that:*
$$f(x) < f(y)$$

*B. In addition, for $K_5$ and $K_6$ positive absolute constants:*

- *for the **Spiked Wishart Model**, for all $x \in \mathcal{B}(x_\star, r_+)$:*
$$\|G(x) - y_\star\|_2 \le K_5 \Big(d^4 \epsilon^{1/2} + \frac{\omega}{\|y_\star\|_2^2}\Big) d^{10} \|y_\star\|_2,$$

- *for the **Spiked Wigner Model**, for all $x \in \mathcal{B}(x_\star, r_+)$:*
$$\|G(x) - y_\star\|_2 \le K_6 \Big(d^4 \epsilon^{1/2} + \frac{\omega}{\|y_\star\|_2^2}\Big) d^{10} \|y_\star\|_2,$$

The previous results imply that, for $\epsilon$ small enough and under the assumptions on the noise level $\omega$, any point $x$ in the benign neighborhood $\mathcal{B}(x_\star, r_+)$ has reconstruction error $\|G(x) - y_\star\|$ which scales optimally according to (5) or (6).

## 3.1   Proofs outline and techniques

The bulk of the analysis will be based on deterministic conditions on the weights of the network. In particular we leverage a set of techinical results recently introduced by [30].

For $W \in \mathbb{R}^{n \times k}$ and $x \in \mathbb{R}^k$, define the operator $W_{+,x} := \mathrm{diag}(Wx > 0)W$ such that $\mathrm{relu}(Wx) = W_{+,x}x$. Moreover let $W_{1,+,x} = (W_1)_{+,x} = \mathrm{diag}(W_1 x > 0)W_1$, and for $2 \le i \le d$:
$$W_{i,+,x} = \mathrm{diag}(W_i, \Pi_{j=i-1}^1 W_{j,+,x} x > 0)W_i,$$
where $\Pi_{i=d}^1 W_i = W_d W_{d-1} \ldots W_1$. Finally we let $\Lambda_x = \Pi_{j=d}^1 W_{j,+,x}$ and note that $G(x) = \Lambda_x x$.

**Definition 1** (Weight Distribution Condition [30]). *We say that $W \in \mathbb{R}^{n \times k}$ satisfies the **Weight Distribution Condition (WDC)** with constant $\epsilon > 0$ if for all $x_1, x_2 \in \mathbb{R}^k$:*

$$\|W_{+,x_1}^{\intercal} W_{+,x_2} - Q_{x_1,x_2}\|_2 \leq \epsilon,$$

*where*

$$Q_{x_1,x_2} = \frac{\pi - \theta_{x_1,x_2}}{2\pi} I_k + \frac{\sin \theta_{x_1,x_2}}{2\pi} M_{\hat{x}_2 \leftrightarrow \hat{x}_2}$$

*and $\theta_{x_1,x_2} = \angle(x_1, x_2)$, $\hat{x}_1 = x_1/\|x_1\|_2$, $\hat{x}_2 = x_2/\|x_2\|_2$, $I_k$ is the $k \times k$ identity matrix and $M_{\hat{x}_1 \leftrightarrow \hat{x}_2}$ is the matrix that sends $\hat{x}_1 \mapsto \hat{x}_2$, $\hat{x}_2 \mapsto \hat{x}_1$, and with kernel span$(\{x_1, x_2\})^{\perp}$.*

Note that $Q_{x_1,x_2}$ is the expected value of $W_{+,x_1}^{\intercal} W_{+,x_2}$ when $W$ has rows $w_i \sim \mathcal{N}(0, I_k/n)$ and if $x_1 = x_2$ then $Q_{x_1,y_2}$ is an isometry up to the scaling factor $1/2$. This condition ensures that the angle between two vectors in the latent space is approximately preserved at the output layer and in turn guarantees the invertibility of the network. Under the Assumption 1, [30] shows that the WDC holds with high probability for all layers of the generative network $G$.

Next we observe that at a differentiable point the gradient of $f$, defined in (4), is given by:

$$\nabla f(x) = \Lambda_x^{\intercal} [\Lambda_x x x^{\intercal} \Lambda_x^{\intercal} - M] \Lambda_x x. \tag{10}$$

Using the WDC, then, we demonstrate that $\nabla_x f(x)$ concentrates up to the noise level around a direction $h_x \in \mathbb{R}^k$ which is a continuous function of nonzero $x$ and $x_\star$. Furthermore using the characterization (9) of the Clarke subdifferential, we show that this concentration extends also at non-differentiable points for subgradients.

A direct analysis then shows that any directional derivative of $f$ at zero is zero and that $h_x$ is small in a neighborhood of $x_\star$ and its negative multiple $-\rho_d x_\star$. This in turn guarantees the existence of a descent direction in the complement of these sets.

Similarly, we use the WDC to show that up to noise level, the loss function $f$ concentrates around:

$$f_E(x) = \frac{1}{4} \left( \frac{1}{2^{2d}} \|x\|_2^4 + \frac{1}{2^{2d}} \|x_\star\|_2^4 - 2\langle x, \tilde{h}_x \rangle^2 \right).$$

where $\tilde{h}_{x,x_\star}$ is continuous for nonzero $x$ and $x_\star$. Directly analyzing the properties of $f_E$ in a neighborhood of $x_\star$ and $-\rho_d x_\star$ allows to derive the first point of Proposition 1. The last part of Proposition 1 follows by noticing that the generator $G$ is locally Lipschitz.

Finally we extend a technique of [31] to control the noise term: in our case a Gaussian Orthogonal matrix $\mathcal{H}$ for the spiked Wigner model (2), and the differene between the empirical and the population covariance $\Sigma_N - \Sigma$ for the spiked Wishart model. The analysis is based on a counting argument on the subspaces spanned by a depth $d$ generative networks, which leads to the sought rate-optimal bounds.

## 4  A subgradient method and numerical experiments

Informed by the analysis in the previous sections, we propose a gradient method for the solution of (4) and verify empirically its optimal properties.

Recall that the main properties of the landscape of the minimization problem (4) were the global minimum in a neighborhood of the true latent vector $x_\star$ and a flat region in correspondence of $-\rho_d x_\star$. Moreover the latter region has larger loss function values than those in the vicinity of $x_\star$. In order to overcome the non-convexity and avoid this bad region we run gradient descent from a random non-zero initialization and its negation. We then pick the iterate which has smaller final loss value.

---

**Algorithm 1** Gradient method for the minizimization problem (4)

---

1: **Input:** Weights $W_i$, observation matrix $M$, step size $\alpha > 0$, number of iterations $T$
2: Choose $\hat{x}_0 \in \mathbb{R}^k \backslash \{0\}$ arbitrary
3: Let $x_0^{(1)} = \hat{x}_0$ and $x_0^{(2)} = -\hat{x}_0$
4: **for** i = 1, 2 **do**
5:     **for** $j = 0, 1, \dots, T-1$ **do**
6:         Compute $v_{x_j}^{(i)} \in \partial f(x_j^{(i)})$
7:         $x_{j+1}^{(i)} \leftarrow x_j^{(i)} - \alpha v_{x_j}^{(i)}$;
8: **if** $f(x_T^{(1)}) < f(x_T^{(2)})$ **then**
9:     Return: $x_T^{(1)}, G(x_T^{(1)})$
10: **else**
11:     Return: $x_T^{(2)}, G(x_T^{(2)})$

---

Note that it will be highly unlikely that the iterates will be at a non-differentiable point, therefore in practice we can consider Algorithm 1 with descent direction $v_x = \nabla f(x)$.

We verify our theoretical claims on synthetic generative priors. We consider 2-layer generative networks with Relu activation functions, hidden layer of dimension $n_1 = 250$, output dimension $n = 1700$ and varying number of latent dimension $k \in [10, 30, 70]$. We randomly sample the weights of the matrix independently from $\mathcal{N}(0, 2/n_i)$, which removes that $2^d$ dependence in Theorem 2. We then consider data $Y$ according the spiked models (1) and (2), where $x_\star \in \mathbb{R}^k$ is chosen so that $y_\star = G(x_\star)$ has unit norm. For the Wishart model we vary the samples $N$ while for the Wigner model we vary the noise level $\nu$ so that the following quantities remain constant for the different networks (latent dimension $k$):

$$\theta_{\mathrm{WS}} := \sqrt{k \log(n_1^2 n)/N}, \qquad \theta_{\mathrm{WG}} := \nu \sqrt{k \log(n_1^2 n)/n}$$

We then plot the reconstruction error given by $\|G(x) - y_\star\|_2$ against $\theta_{\mathrm{WS}} \approx \sqrt{\frac{k}{N}}$ and $\theta_{\mathrm{WG}} \approx \nu\sqrt{\frac{k}{n}}$. As predicted by Theorem 2 the errors scale linearly with respect to these control parameters, and moreover all the plots overlap confirming that these rates are tight with respect to the order of $k$.

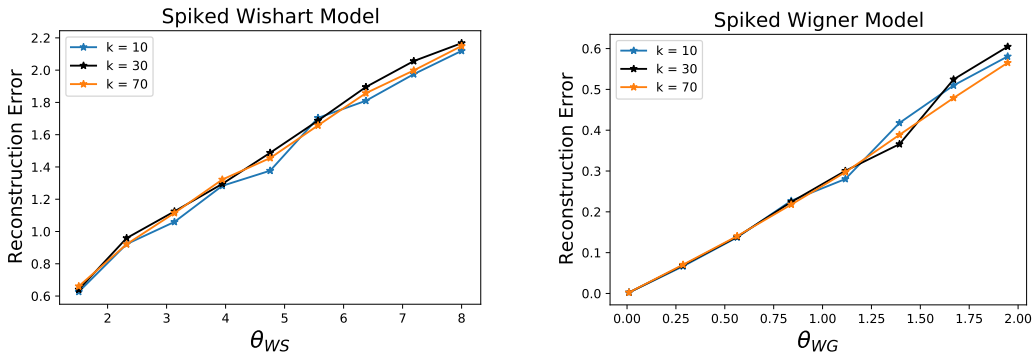

Figure 1: Reconstruction error for the recovery of a spike $y_\star = G(x_\star)$ in Wishart and Wigner model with random generative network priors. Average over 50 random drawing of the network weights and samples. These plots demonstrate that the reconstruction error follow closely our theory.

## Broader Impact

This work promotes the wider use of generative priors in statistical inverse problems. As demonstrated, they can lead to optimal sample-complexity and allow for efficient reconstruction algorithms. The impact of these developments in many applied areas, e.g. medical imaging and diagnostic, will be therefore rather significant as they permit faster measurements acquisition and reduced costs, bringing

a number positive effects including increased accuracy, image quality and potentially scientific discoveries.

As noted by [49], one of the advantages of using a generative priors for inverse problems, is that the measurement operator needs to be known only at test time and only samples from the prior signal distribution are required in order to train the generative network. This means that, once trained, a generative network can be used to solve many different statistical inverse problems.

On the other hand, as [49] observes, the use of generative priors leads to reconstructed images which are highly likely with respect to the empirical distribution that was used for training the generative network. As such they suffer from the same biases of the training data set and can lead to artifacts and hallucinated features.

## Acknowledgments and Disclosure of Funding

PH is supported in part by NSF CAREER Grant DMS-1848087 and NSF Grant DMS-2022205.

## Footnotes

[1]Under the deterministic conditions on the generative network (see below for details), it was shown in [29] that $G$ is invertible and therefore there exists a unique $x_\star$ that satisfies $y_\star = G(x)$.

[2]We write $f(n) \gtrsim g(n)$ if $f(n) \geq Cn$ for some constant $C > 0$ that might depend $\sigma$ and $\|y_\star\|^2$. Similarly for $f(n) \lesssim g(n)$.

[3]The reader is referred to [18] for more details.

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
