[Supplementary Material]

# A  Global landscape analysis under deterministic conditions

As mentioned, the proof of Theorem 2 and Proposition 1, will be based on deterministic conditions on the weights of the network and the noise matrix. In particular we will consider the minimization problem (4) with:

$$M = G(x_\star)G(x_\star)^\mathsf{T} + H,$$

for an unknown symmetric matrix $H \in \mathbb{R}^{n \times n}$ and nonzero $x_\star \in \mathbb{R}^k$.

Recall the definition 1 of the WDC. Below we will say that a $d$-layer generative network $G$ of the form (3), satisfies the WDC with constant $\epsilon > 0$ if every weight matrix $W_i$ has the WDC with constant $\epsilon$ for all $i = 1, \dots d$.

We can now describe the landscape of the minimization problem (4) in a deterministic settings.

**Theorem 3.** *Consider a generative network $G : \mathbb{R}^k \to \mathbb{R}^n$ as in (3) and the minimization problem (4) with unknown nonzero $x_\star$ and symmetric $H$. Fix $\epsilon > 0$ such that $K_1 d^{16} \sqrt{\epsilon} \leq 1$ and let $d \geq 2$. Suppose that $G$ satisfies the WDC with constant $\epsilon$ and assume that:*

$$\|\Lambda_x^\mathsf{T} H \Lambda_x\|_2 \leq \frac{\omega}{2^d} \qquad \text{for all } x \in \mathbb{R}^k, \tag{11}$$

*with $2^d d^{12} w \leq K_2 \|x_\star\|^2$ and $K_2 < 1$.*

*Then for all $x \in \mathbb{R}^k$:*

- *if $x \notin \mathcal{B}(x_\star, r_+) \cup \mathcal{B}(-\rho_d x_\star, r_-) \cup \{0\}$ and $v_x \in \partial f(x)$:*

$$D_{-v_x} f(x) < 0$$

  *where*

$$r_+ = K_3 (d^4 \epsilon^{1/2} + 2^d \omega \|x_\star\|_2^{-2}) d^{10} \|x_\star\|_2,$$

  *and*

$$r_- = K_4 (d^2 \epsilon^{1/4} + 2^{d/2} \omega^{1/2} \|x_\star\|_2^{-1}) d^{10} \|x_\star\|$$

- *if $x \in \mathcal{B}(0, \|x_\star\|_2/16\pi) \backslash \{0\}$ and $v_x \in \partial f(x)$ then*

$$\langle x, v_x \rangle < 0$$

  *while if $x = 0$ and $v \in \mathcal{S}^{k-1}$ then*

$$D_{-v} f(0) = 0$$

*Here $\rho_d$ is a positive number that converges to 1 as $d \to \infty$ and $K_1, \dots, K_4$ are universal constants.*

Similarly, below we give the deterministic version of Proposition 1.

**Proposition 2.** *Under the assumptions of Theorem 3, for any $\phi_d \in [\rho_d, 1]$, it holds that:*

$$f(x) < f(y) \tag{12}$$

*for all $x \in \mathcal{B}(\phi_d x_\star, \varrho \|x_\star\| d^{-12})$ and $y \in \mathcal{B}(-\phi_d x_\star, \varrho \|x_\star\| d^{-12})$ where $\varrho < 1$ is a universal constant.*

The rest of the paper is organized as follows. After summarizing the notation used throughout the paper in Section A.1 and deriving concentration results for the subgradients from the WDC in Section A.2, we give the proof of Theorem 3 in Section A.3. In Section A.4 we prove Proposition 2, while Section A.5 contains the proofs of supplementary lemmas needed in the main results. Finally in Section B, we derive the main Theorem 2 and Proposition 2 from the corresponding deterministic ones by controlling the noise terms and recalling a result of [30] which shows that the WDC holds with high probability.

## A.1 Notation

We now collect the notation that is used throughout the paper. For any real number $a$, let $\text{relu}(a) = \max(a, 0)$ and for any vector $v \in \mathbb{R}^n$, denote the entrywise application of relu as $\text{relu}(v)$. Let $\text{diag}(Wx > 0)$ be the diagonal matrix with $i$-th diagonal element equal to 1 if $(Wx)_i > 0$ and 0 otherwise. For any vector $x$ we denote with $\|x\|$ its Euclidean norm and for any matrix $A$ we denote with $\|A\|$ its spectral norm and with $\|A\|_F$ its Frobenius norm. The euclidean inner product between two vectors $a$ and $b$ is $\langle a, b \rangle$, while for two matrices $A$ and $B$ their Frobenius inner product will be denoted by $\langle A, B \rangle_F$. For any nonzero vector $x \in \mathbb{R}^n$, let $\hat{x} = x/\|x\|$. For a set $S$ we will write $|S|$ for its cardinality and $S^c$ for its complement. Let $\mathcal{B}(x, r)$ be the Euclidean ball of radius $r$ centered at $x$, and $\mathcal{S}^{k-1}$ be the unit sphere in $\mathbb{R}^k$. With $D_v f(x)$ we denote the (normalized) one-sided directional derivative of $f$ in direction $v$: $D_v f(x) = \lim_{t \to 0} \frac{f(x+tv)-f(x)}{t\|v\|_2}$. We will write $\gamma = \Omega(\delta)$ to mean that there exists a positive constant $C$ such that $\gamma \geq C\delta$ and similarly $\gamma = \mathcal{O}(\delta)$ if $\gamma \leq C\delta$. Additionally we will use $a = b + O_1(\delta)$ when $\|a - b\| \leq \delta$, where the norm is understood to be the absolute value for scalars, the Euclidean norm for vectors and the spectral norm for matrices.

## A.2 Preliminaries

At a differentiable point, the gradient of $f$ is given by (10) and will be denoted by $\tilde{v}_x$ and . By the WDC, $\tilde{v}_x$ concentrates up to the noise level around the direction $h_x \in \mathbb{R}^k$:

$$h_x := \big[\frac{1}{2^{2d}}xx^\intercal - \tilde{h}_{x,x_\star}\tilde{h}_{x,x_\star}^\intercal\big]x, \tag{13}$$

where $\tilde{h}_{x,x_\star}$ is defined below and is a continuous function of $x$ and $x_\star$. The vector field $\tilde{h}_{x,x_\star}$ depends on a function that controls how the angles are contracted by the deep network, and defined as:

$$g(\theta) := \cos^{-1}\big(\frac{(\pi - \theta)\cos\theta + \sin\theta}{\pi}\big) \tag{14}$$

With this definition we let $\tilde{h}_{x,x_\star}$ be:

$$\tilde{h}_{x,x_\star} := \frac{1}{2^d}\big[\big(\prod_{i=0}^{d-1}\frac{\pi - \bar{\theta}_i}{\pi}\big)x_\star + \sum_{i=1}^{d-1}\frac{\sin\bar{\theta}_i}{\pi}\big(\prod_{j=i+1}^{d-1}\frac{\pi - \bar{\theta}_j}{\pi}\big)\|x_\star\|\hat{x}\big]$$

where $\theta_i := g(\bar{\theta}_{i-1})$ for $g$ given by (14) and $\theta_0 = \angle(x, y)$. For brevity of notation below we will use $\tilde{h}_x = \tilde{h}_{x,x_\star}$. For later convenience we also define the following vectors:

$$p_x := \Lambda_x^\intercal \Lambda_x x;$$
$$q_x := \Lambda_x^\intercal \Lambda_{x_\star} x_\star;$$
$$\bar{v}_x := \big[p_x p_x^\intercal - q_x q_x^\intercal\big] x;$$
$$\eta_x := \Lambda_x^\intercal H \Lambda_x x.$$

and note that when $f$ is differentiable at $x$, then $\tilde{v}_x := \nabla f(x) = \bar{v}_x - \eta_x$, in particular for zero noise $\tilde{v}_x = \bar{v}_x$.

We now observe the following facts.

**Lemma 1** (Lemma 8 in [29]). *Suppose that $d \geq 2$ and the WDC holds with $\epsilon < 1/(16\pi d^2)^2$, then for all nonzero $x, x_\star \in \mathbb{R}^k$,*

$$\langle \Lambda_x x, \Lambda_{x_\star} x_\star \rangle \geq \frac{1}{4\pi}\frac{1}{2^d}\|x\|_2\|x_\star\|, \tag{15}$$

$$\|\Lambda_x^\intercal \Lambda_{x_\star} x_\star - \tilde{h}_{x,x_\star}\| \leq 24\frac{d^3\sqrt{\epsilon}}{2^d}\|x_\star\|, \text{ and} \tag{16}$$

$$\|\Lambda_x\|^2 \leq \frac{1}{2^d}(1 + 2\epsilon)^d \leq \frac{1 + 4\epsilon d}{2^d} \leq \frac{13}{12}\frac{1}{2^d}. \tag{17}$$

*Proof.* The first two bounds can be found in [29, Lemma 8]. The third bound follows noticing that the WDC implies:

$$\|\Lambda_x\|^2 \leq \Pi_{i=d}^1\|W_{i,+,x}\|^2 \leq \frac{1}{2^d}(1 + 2\epsilon)^d \leq \frac{1 + 4\epsilon d}{2^d} \leq \frac{13}{12}\frac{1}{2^d}$$

where we used $\log(1 + z) \leq z$ and $e^z \leq (1 + 2z)$ for all $0 \leq z \leq 1$. $\qquad\square$

The next lemma shows that the noiseless gradient $\bar{v}_x$, concentrates around $h_x$.

**Lemma 2.** *Suppose $d \geq 2$ and the WDC holds with $\epsilon < 1/(16\pi d^2)^2$, then for all nonzero $x, x_\star \in \mathbb{R}^k$:*

$$\|\bar{v}_x - h_x\| \leq 86 \frac{d^4 \sqrt{\epsilon}}{2^{2d}} \max(\|x_\star\|^2, \|x\|^2)\|x\|$$

We now use the characterization of the Clarke subdifferential given in (9), to derive a bound on the concentration of $v_x \in \partial f(x)$ around $h_x$ up to the noise level.

**Lemma 3.** *Under the assumption of Lemma 2, and with $H$ satisfying (11), for any $v_x \in \partial f(x)$:*

$$\|v_x - h_x\| \leq 86 \frac{d^4 \sqrt{\epsilon}}{2^{2d}} \max(\|x_\star\|^2, \|x\|^2)\|x\| + \frac{\omega}{2^d} \|x\|$$

### A.3 Proof of Theorem 3

We define the set $\mathcal{S}_\beta$ outside which we can lower bound the gradient as:

$$\mathcal{S}_\beta := \left\{ x \in \mathbb{R}^k \mid \|h_x\| \leq \frac{\beta}{2^{2d}} \max(\|x\|^2, \|x_\star\|^2)\|x\| \right\}$$

with:

$$\beta = 5 \cdot \left(86 d^4 \sqrt{\epsilon} + 2^d \omega \|x_\star\|^{-2}\right) \tag{18}$$

Outside the set $\mathcal{S}_\beta$ the gradient is bounded below and the landscape has favorable optimization geometry.

Due to the continuity and piecewise smoothness of the generator $G$ and in turn of the loss function $f$, for any $x, y \neq 0$ there exists a sequence of $\{x_n\} \to x$ such that $f$ is differentiable at each $x_n$ and $D_y f(x) = \lim_{n\to\infty} \nabla f(x_n) \cdot y$. It follows that:

$$D_{-v_x} f(x) = -\lim_{n\to\infty} \tilde{v}_{x_n} \cdot \frac{v_x}{\|v_x\|}$$

as $\nabla f(x_n) = \tilde{v}_{x_n}$. Regarding the right hand side of the above, observe that:

$$
\begin{aligned}
\tilde{v}_{x_n} \cdot v_x &= h_{x_n} \cdot h_x + (v_{x_n} - h_{x_n}) \cdot h_x + h_{x_n} \cdot (v_x - h_x) + (\tilde{v}_{x_n} - h_{x_n}) \cdot (v_x - h_x) \\
&\geq h_{x_n} \cdot h_x - \|\tilde{v}_{x_n} - h_{x_n}\|\|h_x\| - \|h_{x_n}\|\|v_x - h_x\| - \|\tilde{v}_{x_n} - h_{x_n}\|\|v_x - h_x\|, \\
&\geq h_{x_n} \cdot h_x - \frac{86 d^4 \sqrt{\epsilon} + 2^d \omega \|x_\star\|^{-2}}{2^{2d}} \Big( \max(\|x_n\|^2, \|x_\star\|^2)\|x_n\|\|h_x\| + \max(\|x\|^2, \|x_\star\|^2)\|x\|\|h_{x_n}\| \Big) \\
&\quad - \left(\frac{86 d^4 \sqrt{\epsilon} + 2^d \omega \|x_\star\|^{-2}}{2^{2d}}\right)^2 \max(\|x\|^2, \|x_\star\|^2) \max(\|x_n\|^2, \|x_\star\|^2)\|x_n\|\|x\|
\end{aligned}
$$

where the second inequality follows from Lemma 3. Moreover as $h_x$ is continuous in $x$ for all nonzero $x$:

$$
\begin{aligned}
\lim_{n\to\infty} \tilde{v}_{x_n} \cdot v_x &\geq \|h_x\|^2 - 2\frac{86 d^4 \sqrt{\epsilon} + 2^d \omega \|x_\star\|^{-2}}{2^{2d}} \max(\|x\|^2, \|x_\star\|^2)\|x\|\|h_x\| \\
&\quad - \left(\frac{86 d^4 \sqrt{\epsilon} + 2^d \omega \|x_\star\|^{-2}}{2^{2d}}\right)^2 \max(\|x\|^2, \|x_\star\|^2)^2\|x\|^2 \\
&\geq \frac{\|h_x\|}{2} \left[\|h_x\| - 4\frac{86 d^4 \sqrt{\epsilon} + 2^d \omega \|x_\star\|^{-2}}{2^{2d}} \max(\|x\|^2, \|x_\star\|^2)\|x\|\right] \\
&\quad + \frac{1}{2}\left[\|h_x\|^2 - 2\left(\frac{86 d^4 \sqrt{\epsilon} + 2^d \omega \|x_\star\|^{-2}}{2^{2d}}\right)^2 \max(\|x\|^2, \|x_\star\|^2)^2\|x\|^2\right]
\end{aligned}
$$

By our choice of $\beta$ in (18) it follows that for any $x \in S_\beta^c \backslash \{0\}$ :

$$\|h_x\| - 4\frac{86 d^4 \sqrt{\epsilon} + 2^d \omega \|x_\star\|^{-2}}{2^{2d}} \max(\|x\|^2, \|x_\star\|^2)\|x\| \geq \frac{\max(\|x\|^2, \|x_\star\|^2)}{2^{2d}} \|x\| \left(\beta - 4\left(86 d^4 \sqrt{\epsilon} + 2^d \omega \|x_\star\|^{-2}\right)\right),$$

so that:

$$\lim_{n\to\infty} v_{x_n} \cdot v_x \geq \frac{\|h_x\|}{2} \frac{\max(\|x\|^2, \|x_\star\|^2)}{2^{2d}} 86 d^4 \sqrt{\epsilon}\|x\| > 0.$$

The latter equation allows to conclude $D_{-v_x} f(x) < 0$ for any nonzero $x \in S_\beta^c$ and any $v_x \in \partial f(x)$. Finally observe that the radii of the neighborhoods around $x_\star$ and $-\rho_d x_\star$ can be found applying Lemma 4 below with $\beta$ as given in (18).

Next for any nonzero $v \in \mathbb{R}^k$ and $\tau \in \mathbb{R}$ we have:

$$f(\tau v) - f(0) = \frac{\tau^4}{4} \|G(v)G(v)^\mathsf{T}\|_F^2 - \frac{\tau^2}{2} \langle G(v)G(v)^\mathsf{T}, G(x_\star)G(x_\star)^\mathsf{T} + H \rangle_F,$$

which implies that $D_v f(0) = 0$ for any $v \in \mathcal{S}^{k-1}$.

Finally notice that at a differentiable point $x \in \mathbb{R}^k$:

$$\begin{aligned}
\langle \tilde{v}_x, x \rangle &= \langle \Lambda_x^\mathsf{T} [\Lambda_x x x^\mathsf{T} \Lambda_x^\mathsf{T} - \Lambda_{x_\star} x_\star x_\star^\mathsf{T} \Lambda_{x_\star}^\mathsf{T}] \Lambda_x x, x \rangle - \langle \Lambda_x H \Lambda_x, x \rangle \\
&= \|G(x)\|^4 - \langle G(x), G(x_\star) \rangle^2 - \langle \Lambda_x H \Lambda_x, x \rangle \\
&\leq \frac{\|x\|^2}{2^{2d}} \left[ \left( \frac{13}{12} \right)^2 \|x\|^2 - \left( \frac{1}{16\pi^2} - \frac{2^d \omega}{\|x_\star\|^2} \right) \|x_\star\|^2 \right] \\
&\leq \frac{\|x\|^2}{2^{2d}} \left[ 2\|x\|^2 - \frac{1}{32\pi^2} \|x_\star\|^2 \right]
\end{aligned}$$

having used (15), (17) and the assumption on the noise (11) in the first inequality and $2^d d^{12} w \leq K_2 \|x_\star\|^2$ with $d \geq 2$ in the last one. We conclude that if $f$ is differentiable at $x \in \mathcal{B}(0, \|x_\star\|/16\pi)$ then $\langle x, \tilde{v}_x \rangle < 0$.

If $f$ is not differentiable at a nonzero $x \in \mathcal{B}(0, \|x_\star\|/16\pi)$, then by (9) for any $v_x \in \partial f(x)$:

$$\begin{aligned}
\langle v_x, x \rangle &= \langle c_1 v_1 + c_2 v_2 + \cdots + c_T v_T, x \rangle \\
&\leq (c_1 + c_2 \cdots + c_T) \frac{\|x\|^2}{2^{2d}} \left[ 2\|x\|^2 - \frac{1}{32\pi^2} \|x_\star\|^2 \right] < 0
\end{aligned}$$

### A.3.1 Control of the zeros of $h_x$

In this section we show that $h_x$ is nonzero outside two neighborhoods of $x_\star$ and $-\rho_d x_\star$.

**Lemma 4.** *Suppose $8\pi d^6 \sqrt{\beta} \leq 1$. Define:*

$$\rho_d := \sum_{i=0}^{d-1} \frac{\sin \check{\theta}_i}{\pi} \left( \prod_{j=i+1}^{d-1} \frac{\pi - \check{\theta}_j}{\pi} \right)$$

*where $\check{\theta}_0 = \pi$ and $\check{\theta}_i = g(\check{\theta}_{i-1})$. If $x \in \mathcal{S}_\beta$, then we have either:*

$$|\bar{\theta}_0| \leq 32 d^4 \pi \beta \quad and \quad |\|x\|^2 - \|x_\star\|^2| \leq 258\pi\beta d^6 \|x_\star\|$$

*or*

$$|\bar{\theta}_0 - \pi| \leq 8\pi d^4 \sqrt{\beta} \quad and \quad |\|x\|^2 - \rho_d^2 \|x_\star\|^2| \leq 281\pi^2 \sqrt{\beta} d^{10} \|x_\star\|.$$

*In particular, we have:*

$$\mathcal{S}_\beta \subset \mathcal{B}(x_\star, R_1 \beta d^{10} \|x_\star\|) \cup \mathcal{B}(-\rho_d x_\star, R_2 \sqrt{\beta} d^{10} \|x_\star\|)$$

*where $R_1, R_2$ are numerical constants and $\rho_d \to 1$ as $d \to \infty$.*

*Proof.* Without loss of generality, let $x_\star = e_1$ and $x = r \cos \bar{\theta}_0 \cdot e_1 + r \sin \bar{\theta}_0 \cdot e_2$, for some $\bar{\theta}_0 \in [0, \pi]$, and $r \geq 0$. Recall that we call $\hat{x} = x/\|x\|$ and $\hat{x}_\star = x_\star/\|x_\star\|$. We then introduce the following notation:

$$\xi = \prod_{i=0}^{d-1} \frac{\pi - \bar{\theta}_i}{\pi}, \quad \zeta = \sum_{i=0}^{d-1} \frac{\sin \bar{\theta}_i}{\pi} \prod_{j=i+1}^{d-1} \frac{\pi - \bar{\theta}_j}{\pi}, \quad r = \|x\|, \quad R = \max(r^2, 1), \quad (19)$$

where $\theta_i = g(\bar{\theta}_{i-1})$ with $g$ as in (14), and observe that $2^d \tilde{h}_x = (\xi \hat{x}_\star + \zeta \hat{x})$. Let $\alpha := 2^d \langle \tilde{h}_x, \hat{x} \rangle$, then we can write:

$$h_x = \left[ \frac{\langle x, x \rangle}{2^{2d}} x - \langle \tilde{h}_x, x \rangle \tilde{h}_x \right] = \frac{r}{2^{2d}} \left[ r^2 \hat{x} - \alpha (\xi \hat{x}_\star + \zeta \hat{x}) \right].$$

Using the definition of $\hat{x}$ and $\hat{x}_\star$ we obtain:

$$\frac{2^{2d}h_x}{r} = \left[(r^2 - \alpha\,\zeta)\cos\bar\theta_0 - \alpha\,\xi\right]\cdot e_1 + \left[r^2 - \alpha\zeta\right]\sin\bar\theta_0 \cdot e_2,$$

and conclude that since $x \in \mathcal{S}_\beta$, then:

$$|(r^2 - \alpha\,\zeta)\cos\bar\theta_0 - \alpha\,\xi| \leq \beta R \tag{20}$$
$$|[r^2 - \alpha\zeta]\sin\bar\theta_0| \leq \beta R. \tag{21}$$

We now list some bounds that will be useful in the subsequent analysis. We have:

$$\bar\theta_i \leq \bar\theta_{i-1} \ \text{ for } \ i \geq 1 \tag{22}$$
$$\bar\theta_i \leq \cos^{-1}(1/\pi) \ \text{ for } \ i \geq 2 \tag{23}$$
$$|\xi| \leq 1 \tag{24}$$
$$|\zeta| \leq \frac{d}{\pi}\sin\bar\theta_0 \tag{25}$$
$$\xi \geq \frac{\pi - \bar\theta_0}{\pi}d^{-3} \tag{26}$$
$$\check\theta_i \leq \frac{3\pi}{i+3} \ \text{ for } \ i \geq 0 \tag{27}$$
$$\check\theta_i \geq \frac{\pi}{i+1} \ \text{ for } \ i \geq 0 \tag{28}$$
$$\bar\theta_0 = \pi + O_1(\delta) \Rightarrow |\xi| \leq \frac{\delta}{\pi} \tag{29}$$
$$\bar\theta_0 = \pi + O_1(\delta) \Rightarrow \zeta = \rho_d + O_1(3d^3\delta) \text{ if } \frac{d^2\delta}{\pi} \leq 1 \tag{30}$$
$$1/\pi \leq \alpha \leq 1. \tag{31}$$

The identities (22) through (30) can be found in Lemma 16 of [31], while the identity (31) follows by noticing that $\alpha = \xi\cos\bar\theta_0 + \zeta = \cos\theta_d$ and using (23) together with $d \geq 2$.

**Bound on $R$.** We now show that if $x \in \mathcal{S}_\beta$, then $r^2 \leq 4d$ and therefore $R \leq 4d$.

If $r^2 \leq 1$, then the claim is trivial. Take $r^2 > 1$, then note that either $|\sin\bar\theta_0| \geq 1/\sqrt{2}$ or $|\cos\bar\theta_0| \geq 1/\sqrt{2}$ must hold. If $|\sin\bar\theta_0| \geq 1/\sqrt{2}$ then from (21) it follows that $r^2 - \alpha\zeta \leq \sqrt{2}\beta R = \sqrt{2}\beta r^2$ which implies:

$$r^2 \leq \frac{\alpha\,\zeta}{1 - \sqrt{2}\beta} \leq \frac{1}{(1 - \sqrt{2}\beta)}\frac{d}{\pi} \leq \frac{d}{2}$$

using (25) and (31) in the second inequality and $\beta < 1/4$ in the third. Next take $|\cos\bar\theta_0| \geq 1/\sqrt{2}$, then (20) implies $|r^2 - \alpha\zeta| \leq \sqrt{2}(\beta r^2 + \alpha\xi)$ which in turn results in:

$$r^2 \leq \frac{\alpha(\zeta + \sqrt{2}\xi)}{1 - \sqrt{2}\beta} \leq 4d$$

using (24), (25), (31) and $\beta < 1/4$. In conclusion if $x \in \mathcal{S}_\beta$ then $r^2 \leq 4d \Rightarrow R \leq 4d$.

**Bounds on $\bar\theta_0$.** We now show we only have to analyze the small angle case $\bar\theta_0 \approx 0$ and the large angle case $\bar\theta_0 \approx \pi$.

At least one of the following three cases must hold:

1. $\sin\bar\theta_0 \leq 16\beta\pi d^4$: Then we have $\bar\theta = O_1(32\pi\beta\pi d^4)$ or $\bar\theta = \pi + O_1(32\pi\beta\pi d^4)$ as $32\pi\beta\pi d^4 < 1$.

2. $|r^2 - \alpha\zeta| < \sqrt{\beta}R$: Then (20), (31) and $\beta < 1$ yield $|\xi| \leq 2\sqrt{\beta}\pi R$. Using (26), we then get
$$\bar\theta = \pi + O_1(2\sqrt{\beta}\pi^2 d^3 R).$$

3. $\sin \bar{\theta}_0 > 16\beta\pi d^4$ and $|r^2 - \alpha\zeta| \geq \sqrt{\beta}R$: Then (21) gives $|r^2 - \alpha\zeta| \leq \beta M / \sin\bar{\theta}_0$ which used with (20) leads to:

$$|\alpha\xi| \leq \beta R + |r^2 - \alpha\zeta| \leq \beta R + \frac{\beta R}{\sin\bar{\theta}_0} \leq 2\frac{\beta R}{\sin\bar{\theta}_0}.$$

Then using (31), the assumption on $\sin\bar{\theta}_0$ and $R \leq 4d$ we obtain $\xi \leq d^{-3}/2$. The latter together with (26) leads to $\bar{\theta}_0 \geq \pi/2$. Finally as $|r^2 - \alpha\zeta| \geq \sqrt{\beta}R$ then (21) leads to $|\sin\bar{\theta}_0| \leq \sqrt{\beta}$. Therefore as $\bar{\theta}_0 \geq \pi/2$ and $\beta < 1$, we can conclude that $\bar{\theta}_0 = \pi + O_1(2\sqrt{\beta})$.

Inspecting the three cases, and recalling that $R \leq 4d$, we can see that it suffices to analyze the small angle case $\bar{\theta}_0 = O_1(32d^4\pi\beta)$ and the large angle case $\bar{\theta} = \pi + O_1(8\sqrt{\beta}\pi^2 d^4)$.

**Small angle case.** We assume $\bar{\theta}_0 = O_1(\delta)$ with $\delta = 32d^4\pi\beta$ and show that $\|x\|^2 \approx \|x_\star\|^2$.

We begin collecting some bounds. Since $\bar{\theta}_i \leq \bar{\theta}_0 \leq \delta$, then $1 \geq \xi \geq (1 - \delta/\pi)^d \geq 1 + O_1(2d\delta/\pi)$ assuming $\delta d/\pi \leq 1/2$, which holds true since $64d^5\beta < 1$. Moreover from (25) we have $\zeta = O_1(d\delta/\pi)$. Finally observe that $\cos\bar{\theta}_0 = 1 + O_1(\bar{\theta}_0^2/2) = 1 + O_1(\delta/2)$ for $\delta < 1$. We then have $\alpha = 1 + O_1(2d\delta)$ so that $\alpha\zeta = O_1(d^2\delta)$ and $\alpha\xi = 1 + O_1(4d^2\delta)$. We can therefore rewrite (20) as:

$$(r^2 + O_1(d^2\delta))(1 + O_1(\delta/2)) - (1 + O_1(4d^2\delta)) = O_1(\beta R).$$

Using the bound $r^2 \leq R \leq 4d$ and the definition of $\delta$, we obtain:

$$\begin{aligned} r^2 - 1 &= O_1\Big(\frac{\delta r^2}{2} + d^2\delta + \frac{d^2\delta^2}{2} + 4d^2\delta + 4d\beta\Big) \\ &= O_1(8d^2\delta + 4d\beta) \\ &= O_1(258\pi d^6\beta) \end{aligned} \tag{32}$$

**Large angle case.** Here we assume $\bar{\theta} = \pi + O_1(\delta)$ with $\delta = 8\sqrt{\beta}\pi^2 d^4$ and show that it must be $\|x\|^2 \approx \rho_d^2\|x_\star\|^2$.

From (29) we know that $\xi = O_1(\delta/\pi)$, while from (30) we know that $\zeta = \rho_d + O_1(3d^3\delta)$ as long as $8\sqrt{\beta}\pi d^6 \leq 1$. Moreover for large angles and $\delta < 1$, it holds $\cos\bar{\theta}_0 = -1 + O_1((\bar{\theta}_0 - \pi)^2/2) = -1 + O_1(\delta^2/2)$. These bounds lead to:

$$\begin{aligned} \alpha &= \xi\cos\bar{\theta}_0 + \zeta \\ &= \rho_d + O_1\Big(\frac{\delta}{\pi} + \frac{\delta^3}{2\pi} + 3d^3\delta\Big) \\ &= \rho_d + O_1(4d^3\delta), \end{aligned}$$

and using $\rho_d \leq d$:

$$\alpha\zeta = \rho_d^2 + O_1(4d^3\delta\rho_d + 3d^3\delta\rho_d + 12d^6\delta) = \rho_d^2 + O_1(20d^6\delta),$$
$$\alpha\xi = O_1\Big(\frac{\delta}{\pi}\rho_d + 4\frac{d^3\delta^2}{\pi}\Big) = O_1(2d^3\delta).$$

Then recall that (20) is equivalent to $(r^2 - \alpha\zeta)\cos\bar{\theta}_0 - \alpha\xi = O_1(4\beta d)$, that is:

$$\big(r^2 - \rho_d^2 + O_1(20d^6\delta)\big)\big(1 + O_1(\delta^2/2)\big) + O_1(2d^3\delta) = O_1(4\beta d)$$

and in particular:

$$\begin{aligned} r^2 - \rho_d^2 &= O_1\Big(20d^6\delta + 10d^6\delta^3 + \frac{\rho_d\delta^2}{2} + \frac{r^2\delta^2}{2} + 2d^3\delta + 4\beta d\Big) \\ &= O_1\big(35d^6\delta + 4\beta d\big) \\ &= O_1(281\pi^2\sqrt{\beta}d^{10}) \end{aligned} \tag{33}$$

where we used $\rho_d \leq d$, the definition of $\delta$ and $\delta < 1$.

**Controlling the distance.** We have shown that it is either $\bar{\theta}_0 \approx 0$ and $\|x\|^2 \approx \|x_\star\|^2$ or $\bar{\theta}_0 \approx \pi$ and $\|x\|^2 \approx \rho_d^2 \|x_\star\|^2$. We can therefore conclude that it must be either $x \approx x_\star$ or $x \approx -\rho_d x_\star$.

Observe that if a two dimensional point is known to have magnitude within $\Delta r$ of some $r$ and is known to be within an angle $\Delta\theta$ from 0, then its Euclidean distance to the point of coordinates $(r, 0)$ is no more that $\Delta r + (r + \Delta r)\Delta\theta$. Similarly we can write:

$$\|x - x_\star\| \le |\|x\| - \|x_\star\|| + (\|x_\star\| + |\|x\| - \|x_\star\||)\bar{\theta}_0. \tag{34}$$

In the small angle case, by (32), (34), and $\|x_\star\| \, |\|x\| - \|x_\star\|| \le |\|x\|^2 - \|x_\star\|^2|$, we have:

$$\|x - x_\star\| \le 258\pi d^6\beta + (1 + 258\pi d^6\beta)\, 32 d^4 \pi\beta \le 550\,\pi d^{10}\beta.$$

Next we notice that $\rho_2 = 1/\pi$ and $\rho_d \ge \rho_{d-1}$ as follows from the definition and (27), (28). Then considering the large angle case and using (33) we have:

$$|\|x\| - \rho_d| \le \frac{281\pi^2\sqrt{\beta}d^{10}}{\|x\| + \rho_d} \le 281\pi^3\sqrt{\beta}d^{10}.$$

The latter, together with (34), yields:

$$\begin{aligned}\|x + \rho_d x_\star\| &\le |\|x\| - \rho_d| + (\rho_d + |\|x\| - \rho_d|)(\pi - \bar{\theta}_0)\\ &\le 281\pi^3\sqrt{\beta}d^{10} + (d + 281\pi^3\sqrt{\beta}d^{10})8\sqrt{\beta}\pi^2 d^4\\ &\le 284\pi^3\sqrt{\beta}d^{10}\end{aligned}$$

where in the second inequality we have used $\rho_d \le d$ and in the third $8\sqrt{\beta}\pi d^6 \le 1$.

We conclude by noticing that $\rho_d \to 1$ as $d \to 1$ as shown in [31, Lemma 16]. $\qquad\square$

### A.4 Proof of Proposition 2

Recall that $f(x) := 1/4\|G(x)G(x)^\intercal - G(x_\star)G(x_\star)^\intercal - H\|_F^2$, we next define the following loss functions:

$$f_0(x) := \frac{1}{4}\|G(x)G(x)^\intercal - G(x_\star)G(x_\star)^\intercal\|_F^2,$$

$$f_H(x) := f_0(x) - \frac{1}{2}\langle G(x)G(x)^\intercal - G(x_\star)G(x_\star)^\intercal, H\rangle_F,$$

$$f_E(x) := \frac{1}{4}\left(\frac{1}{2^{2d}}\|x\|^4 + \frac{1}{2^{2d}}\|x_\star\|^4 - 2\langle x, \tilde{h}_x\rangle^2\right).$$

In particular notice that $f(x) = f_H(x) + 1/4\|H\|_F^2$. Below we show that assuming the WDC is satisfied $f_0(x)$ concentrates around $f_E(x)$.

**Lemma 5.** *Suppose that $d \ge 2$ and the WDC holds with $\epsilon < 1/(16\pi d^2)^2$, then for all nonzero $x, x_\star \in \mathbb{R}^k$*

$$|f_0(x) - f_E(x)| \le \frac{16}{2^{2d}}(\|x\|^4 + \|x_\star\|^4)d^4\sqrt{\epsilon}$$

We next consider the loss $f_E$ and show that in a neighborhood $-\rho_d x_\star$, this loss function has larger values than in a neighborhood of $x_\star$.

**Lemma 6.** *Fix $0 < a \le 1/(2\pi^3 d^3)$ and $\phi_d \in [\rho_d, 1]$ then:*

$$f_E(x) \le \frac{1}{2^{2d+2}}\|x_\star\|^4 + \frac{1}{2^{2d+2}}\left[(a + \phi_d)^4 - 2\phi_d^2 + 2\pi da\right]\|x_\star\|^4 \quad \forall x \in \mathcal{B}(\phi_d x_\star, a\|x_\star\|) \text{ and}$$

$$f_E(x) \ge \frac{1}{2^{2d+2}}\|x_\star\|^4 + \frac{1}{2^{2d+2}}\left[(a - \phi_d)^4 - 2\rho_d^2\phi_d^2 - 40\pi d^3 a\right]\|x_\star\|^4 \quad \forall x \in \mathcal{B}(-\phi_d x_\star, a\|x_\star\|).$$

The above two lemmas are now used to prove Proposition 2.

*Proof of Proposition 2.* Let $x \in \mathcal{B}(\pm \phi_d x_\star, \varphi \|x_\star\|)$ for a $0 < \varphi < 1$ that will be specified below, and observe that by the assumptions on the noise:

$$|\langle G(x)G(x)^\mathsf{T} - G(x_\star)G(x_\star)^\mathsf{T}, H\rangle_F| \le |G(x)^\mathsf{T}HG(x)| + |G(x_\star)^\mathsf{T}HG(x_\star)|$$
$$\le \frac{\omega}{2^d}(\|x\|^2 + \|x_\star\|^2)$$
$$\le \frac{\omega}{2^d}((\phi_d + \varphi)^2 + 1)\|x_\star\|^2,$$

and therefore by Lemma 5:

$$|f_0(x) - f_E(x)| + \frac{1}{2}|\langle G(x)G(x)^\mathsf{T} - G(x_\star)G(x_\star)^\mathsf{T}, H\rangle_F| \le$$
$$\le \frac{16}{2^{2d}}((\phi_d + \varphi)^4 + 1)\|x_\star\|^4 d^4 \sqrt{\epsilon} + \frac{\omega}{2^d}((\phi_d + \varphi)^2 + 1)\|x_\star\|^2$$
$$\le \frac{272}{2^{2d}}\|x_\star\|^4 d^4 \sqrt{\epsilon} + \frac{\omega}{2^d}((\phi_d + \varphi)^2 + 1)\|x_\star\|^2$$

We next take $\varphi = \epsilon$ and $x \in \mathcal{B}(\phi_d x_\star, \varphi\|x_\star\|)$, so that by Lemma 6 and the assumption $2^d d^{12} w \le K_2 \|x_\star\|^2$, we have:

$$f_H(x) \le f_E(x) + |f_0(x) - f_E(x)| + \frac{1}{2}|\langle G(x)G(x)^\mathsf{T} - G(x_\star)G(x_\star)^\mathsf{T}, H\rangle_F|$$
$$\le \frac{1}{2^{2d+2}}\left[1 + (\epsilon + \phi_d)^4 - 2\phi_d^2 + 2\pi d\epsilon\right]\|x_\star\|^4 + 272 d^4 \sqrt{\epsilon}\|x_\star\|^4 + \frac{\omega}{2^{d+1}}(2 + 2\epsilon + \epsilon^2)\|x_\star\|^2$$
$$\le \frac{1}{2^{2d+2}}\left[1 - 2\phi_d^2 + (\epsilon + \phi_d)^4\right]\|x_\star\|^4 + \frac{1}{2^{2d}}\left(\frac{3}{2}2^d\|x_\star\|^{-2}\omega + \frac{\pi d}{2} + 272 d^4\right)\sqrt{\epsilon}\|x_\star\|^4 + \frac{\omega}{2^d}\|x_\star\|^2$$
$$\le \frac{1}{2^{2d+2}}\left[1 - 2\phi_d^2 + (\epsilon + \phi_d)^4\right]\|x_\star\|^4 + \frac{1}{2^{2d}}\left(\frac{3}{2}K_2 d^{-12} + \frac{\pi d}{2} + 272 d^4\right)\sqrt{\epsilon}\|x_\star\|^4 + K_2 \frac{\|x_\star\|^4}{2^{2d}}d^{-12}.$$

Similarly if $y \in \mathcal{B}(-\phi_d x_\star, \varphi\|x_\star\|)$, and $\varphi = \epsilon$ we obtain:

$$f_H(y) \ge f_E(y) - |f_0(y) - f_E(y)| - \frac{1}{2}|\langle G(y)G(y)^\mathsf{T} - G(x_\star)G(x_\star)^\mathsf{T}, H\rangle|$$
$$\ge \frac{1}{2^{2d+2}}\left[1 - 2\phi_d^2 \rho_d^2 + (\epsilon - \phi_d)^4\right]\|x_\star\|^4 - \frac{1}{2^{2d}}\left(\frac{3}{2}2^d\|x_\star\|^{-2}\omega + 10\pi d^3 + 272 d^4\right)\sqrt{\epsilon}\|x_\star\|^4 - \frac{\omega}{2^d}\|x_\star\|^2$$
$$\ge \frac{1}{2^{2d+2}}\left[1 - 2\phi_d^2 \rho_d^2 + (\epsilon - \phi_d)^4\right]\|x_\star\|^4 - \frac{1}{2^{2d}}\left(\frac{3}{2}K_2 d^{-12} + 10\pi d^3 + 272 d^4\right)\sqrt{\epsilon}\|x_\star\|^4 - K_2 \frac{\|x_\star\|^4}{2^{2d}}d^{-12}.$$

In order to guarantee that $f(y) > f(x)$, it suffices to have:

$$2(1 - \rho_d^2)\phi_d^2 - 8K_2 d^{-12} > 4C_d\sqrt{\epsilon}$$

with $C_d := (544 d^4 + 10\pi d^3 \pi + 3K_2 d^{-12} + \pi d/2 + 1/100)$, that is to require:

$$\varphi = \epsilon < \left(\frac{(1 - \rho_d^2)\phi_d^2 - 4K_2 d^{-12}}{2C_d}\right)^2.$$

Finally notice that by Lemma 17 in [31] it holds that $1 - \rho_d \ge (K(d + 2))^{-2}$ for some numerical constant $K$, we therefore choose $\epsilon = \varrho/d^{12}$ for some $\varrho > 0$ small enough. $\qquad\square$

## A.5 Supplementary proofs

Below we prove Lemma 2 on the concentration of the gradient of $f$ at a differentiable point.

*Proof of Lemma 2.* We begin by noticing that:

$$\bar{v}_x - h_x = \left[\langle p_x, x\rangle p_x - \langle x, x\rangle \frac{x}{2^{2d}}\right] + \left[\langle \tilde{h}_x, x\rangle x - \langle q_x, x\rangle x\right].$$

Below we show that:

$$\|\langle p_x, x\rangle p_x - \langle x, x\rangle \frac{x}{2^{2d}}\| \le \frac{50}{2^{2d}}d^3\sqrt{\epsilon}\max\{\|x\|^2, \|x_\star\|^2\}\|x\|. \tag{35}$$

and

$$\|\langle q_x, x \rangle p_x - \langle \tilde{h}_x, x \rangle \tilde{h}_x | \leq \frac{36}{2^{2d}} d^4 \sqrt{\epsilon} \max\{\|x\|^2, \|x_\star\|^2\}\|x\|. \tag{36}$$

from which the thesis follows.

Regarding equation (35) observe that:

$$\|\langle p_x, x \rangle p_x - \langle x, x \rangle \frac{x}{2^{2d}}\| = \|\langle p_x, x \rangle \big[ p_x - \frac{x}{2^d} \big] + \langle p_x - \frac{x}{2^d}, \frac{x}{2^d} \rangle x\|$$

$$\leq \big( \|\Lambda_x x\|^2 + \frac{\|x\|^2}{2^d} \big) \|p_x - \frac{x}{2^d}\|$$

$$\leq \frac{50}{2^{2d}} d^3 \sqrt{\epsilon} \|x\|^3$$

where in the first inequality we used $\langle p_x, x \rangle = \|\Lambda_x x\|^2$ and in the second we used equations (16) and (17) of Lemma 1.

Next note that:

$$\|\langle q_x, x \rangle q_x - \langle \tilde{h}_x, x \rangle \tilde{h}_x\| = \|\langle q_x, x \rangle (q_x - \tilde{h}_x) + \langle q_x - \tilde{h}_x, x \rangle \tilde{h}_x\|$$

$$\leq (\|q_x\| + \|\tilde{h}_x\|) \|x\| \|q_x - \tilde{h}_x\|$$

$$\leq \big( \frac{13}{12} + 1 + \frac{d}{\pi} \big) \frac{\|x\| \|x_\star\|}{2^d} \|q_x - \tilde{h}_x\|$$

$$\leq \frac{3}{2} d \frac{\|x\| \|x_\star\|}{2^d} \|q_x - \tilde{h}_x\|$$

where in the second inequality we have the bound (17) and the definition of $\tilde{h}_x$. Equation (36) is then found by appealing to equation (16) in Lemma 1. $\qquad\square$

The previous lemma is now used to control the concentration of the subgradients $v_x$ of $f$ around $h_x$.

*Proof of Lemma 3.* When $f$ is differentiable at $x$, $\nabla f(x) = \tilde{v}_x = \bar{v}_x + \eta_x$, so that by Lemma 2 and the assumption on the noise:

$$\|v_x - h_x\| \leq \|\bar{v}_x - h_x\| + \|\eta_x\|$$

$$\leq 86 \frac{d^4 \sqrt{\epsilon}}{2^{2d}} \max(\|x_\star\|^2, \|x\|^2)\|x\| + \frac{\omega}{2^d} \|x\|. \tag{37}$$

Observe, now, that by (9), for any $x \in \mathbb{R}^k$, $v_x \in \partial f(x) = \mathrm{conv}(v_1, \ldots, v_t)$, and therefore $v_x = a_1 v_1 + \cdots + a_T v_T$ for some $a_1, \ldots, a_T \geq 0$, $\sum_i a_i = 1$. Moreover for each $v_i$ there exist a $w_i$ such that $v_i = \lim_{\delta \to 0^+} \tilde{v}_{x+\delta w_i}$. Therefore using equation (37), the continuity of $h_x$ with respect to nonzero $x$ and $\sum_i a_i = 1$:

$$\|v_x - h_x\| \leq \sum_{i=1}^T a_i \|v_i - h_x\|$$

$$\leq \sum_{i=1}^T a_i \lim_{\delta \to 0} \|\tilde{v}_{x+\delta w_i} - h_{x+\delta w_i}\|$$

$$\leq 86 \frac{d^4 \sqrt{\epsilon}}{2^{2d}} \max(\|x_\star\|^2, \|x\|^2)\|x\| + \frac{\omega}{2^d} \|x\|.$$

$\qquad\square$

We now prove Lemma 5 on the concentration of the noiseless objective function.

*Proof of Lemma 5.* Observe that:

$$|f_0(x) - f_E(x)| \leq \frac{1}{4} |\|G(x)\|^4 - \frac{1}{2^{2d}} \|x\|^4|$$

$$+ \frac{1}{4} |\|G(x_\star)\|^4 - \frac{1}{2^{2d}} \|x_\star\|^4|$$

$$+ \frac{1}{2} |\langle G(x), G(x_\star) \rangle^2 - \langle x, \tilde{h}_x \rangle|.$$

We analyze each term separately. The first term can be bounded as:

$$\frac{1}{4}\big|\|G(x)\|^4 - \frac{1}{2^{2d}}\|x\|^4\big| = \frac{1}{4}\big|\|G(x)\|^2 + \frac{1}{2^d}\|x\|^2\big|\ \big|\|G(x)\|^2 - \frac{1}{2^d}\|x\|^2\big|$$

$$\leq \frac{1}{4}\frac{1}{2^d}\big(\frac{13}{12}+1\big)\|x\|^2\ \big|\|G(x)\|^2 - \frac{1}{2^d}\|x\|^2\big|$$

$$\leq \frac{1}{4}\frac{1}{2^d}\big(\frac{13}{12}+1\big)\|x\|^2\ 24\frac{d^3\sqrt{\epsilon}}{2^d}\|x\|^2$$

$$\leq \frac{1}{2^{2d}}13d^3\ \sqrt{\epsilon}\|x\|^4$$

where in the first inequality we used (17) and in the second inequality (16) . Similarly we can bound the second term:

$$\frac{1}{4}\big|\|G(x_\star)\|^4 - \frac{1}{2^{2d}}\|x_\star\|^4\big| \leq \frac{1}{2^{2d}}13d^3\ \sqrt{\epsilon}\|x_\star\|^4.$$

We next note that $\|\tilde{h}_x\| \leq 2^{-d}(1+d/\pi)\|x_\star\|$ and therefore from (17) and $d \geq 2$ we obtain:

$$\big|\|G(x)\|\|G(x_\star)\| + \|x\|\|\tilde{h}_x\|\big| \leq \frac{1}{2^d}\big(\frac{13}{12}+1+\frac{d}{\pi}\big)\|x\|\|x_\star\| \leq \frac{1}{2^d}\frac{3}{2}d\|x\|\|x_\star\|$$

We can then conclude that:

$$\frac{1}{2}\big|\langle G(x), G(x_\star)\rangle^2 - \langle x, \tilde{h}_x\rangle^2\big| \leq \frac{1}{2}\big|\langle x, \Lambda_x^T\Lambda_{x_\star}x_\star - \tilde{h}_x\rangle\big|\ \big|\|G(x)\|\|G(x_\star)\| + \|x\|\|\tilde{h}_x\|\big|$$

$$\leq \frac{1}{2}\|x\|24\frac{d^3\sqrt{\epsilon}}{2^d}\|x_\star\|\ \frac{1}{2^d}\frac{3}{2}d\ \|x\|\|x_\star\|$$

$$\leq \frac{9}{2^{2d}}d^4\sqrt{\epsilon}(\|x_\star\|^4 + \|x\|^4) \qquad\qquad .$$

$\square$

Below we prove lower and upper bound on the loss $f_E$ as in Lemma 6.

*Proof of Lemma 6.* Let $x \in \mathcal{B}(\phi_d x_\star, a\|x_\star\|)$ then observe that $0 \leq \bar{\theta}_i \leq \bar{\theta}_0 \leq \pi a/2\phi_d$ and $(\phi_d - a)\|x_\star\| \leq \|x\| \leq (a+\phi_d)\|x_\star\|$. Then observe that:

$$\langle x, \tilde{h}_d\rangle = \frac{1}{2^d}\big(\prod_{i=0}^{d-1}\frac{\pi-\bar{\theta}_i}{\pi}\big)\|x_\star\|\|x\|\cos\bar{\theta}_0 + \frac{1}{2^d}\sum_{i=0}^{d-1}\frac{\sin\bar{\theta}_i}{\pi}\prod_{j=i+1}^{d-1}\frac{\pi-\bar{\theta}_j}{\pi}\ \|x_\star\|\|x\|$$

$$\geq \frac{1}{2^d}\big(\prod_{i=0}^{d-1}\frac{\pi-\frac{\pi a}{2\phi_d}}{\pi}\big)\ (\phi_d-a)\|x_\star\|^2\big(1-\frac{\pi^2a^2}{8\phi_d^2}\big)$$

$$\geq \frac{1}{2^d}\big(1-\frac{da}{\phi_d}\big)(\phi_d-a)\big(1-\frac{\pi^2a^2}{8\phi_d^2}\big)\|x_\star\|^2.$$

using $\cos\theta \geq 1 - \theta^2/2$ and $(1-x)^d \geq (1-2dx)$ as long as $0 \leq x \leq 1$. We can therefore write:

$$f_E(x) - \frac{\|x_\star\|^4}{2^{2d+2}} \leq \frac{1}{2^{2d+2}}\|x\|^4 - \frac{1}{2^{2d+1}}\big(1-\frac{da}{\phi_d}\big)^2(\phi_d-a)^2\big(1-\frac{\pi^2a^2}{8\phi_d^2}\big)^2\|x_\star\|^4$$

$$\leq \frac{1}{2^{2d+2}}\Big[(\phi_d+a)^4 - 2\big(1-2\frac{da}{\phi_d}\big)(\phi_d-a)^2\big(1-\frac{\pi^2a^2}{4\phi_d^2}\big)\Big]\|x_\star\|^4$$

where in the second inequality we used $(1-x)^2 \geq 1-2x$ for all $x \in \mathbb{R}$. We then observe that:

$$\big(1-2\frac{da}{\phi_d}\big)(\phi_d-a)^2\big(1-\frac{\pi^2a^2}{4\phi_d^2}\big) \geq \big(1-\frac{\pi^2a^2}{4\phi_d^2}-\frac{2ad}{\phi_d}\big)\phi_d^2 + a(a-2\phi_d)\big(1-2\frac{da}{\phi_d}\big)\big(1-\frac{\pi^2a^2}{4\phi_d^2}\big)$$

$$\geq \phi_d^2 - a\big(\frac{1}{2\pi d^3}+2d\phi_d\big) + a(a-2\phi_d)\big(1-2\frac{da}{\phi_d}\big)\big(1-\frac{\pi^2a^2}{4\phi_d^2}\big)$$

$$\geq \phi_d^2 - a\big(\frac{1}{2\pi d^3}+2d\phi_d+2\phi_d\big)$$

$$\geq \phi_d^2 - \pi da,$$

where in the second inequality we have used $\pi^3 d^3 a \leq 2$ and in the last one $d \geq 2$ and $\phi_d \leq 1$. We can then conclude that:

$$f_E(x) - \frac{\|x_\star\|^4}{2^{2d+2}} \leq \frac{1}{2^{2d+2}} \left[ (\phi_d + a)^4 - 2(\phi_d^2 - \pi da) \right] \|x_\star\|^4$$

We next take $x \in \mathcal{B}(-\phi_d x_\star, a\|x_\star\|)$ which implies $0 \leq \pi - \bar{\theta}_0 \leq \pi^2 a/2 =: \delta$ and $\|x\| \leq (a + \phi_d)\|x_\star\|$. We then note that for $\xi$ and $\zeta$ as defined in (19) we have:

$$\begin{aligned} |2^d x^\intercal \tilde{h}_x|^2 &\leq (|\xi| + |\zeta|)^2 (a + \phi_d)^2 \|x_\star\|^4 \\ &\leq \left( \frac{\delta}{\pi} + 3d^3\delta + \rho_d \right)^2 (a + \phi_d)^2 \|x_\star\|^4 \\ &\leq \left( \frac{\pi^3 d^3}{2} a + \rho_d \right)^2 (a + \phi)^2 \|x_\star\|^4 \\ &\leq (2\pi^3 d^3 a + \rho_d^2)(a + \phi_d)^2 \|x_\star\|^4 \\ &\leq 20\pi d^3 a + \rho_d^2 \phi_d^2 \end{aligned}$$

where the second inequality is due to (29) and (30), the rest from $d \geq 2$, $\rho_d \leq \phi_d \leq 1$ and $2\pi^3 d^3 a \leq 1$. Finally using $(\phi_d - a)\|x_\star\| \leq \|x\|$, we can then conclude that:

$$f_E(x) - \frac{\|x_\star\|^4}{2^{2d+2}} \geq \frac{1}{2^{2d+2}} \left[ (\phi_d - a)^4 - 2(20\pi d^3 a + \rho_d^2 \phi_d^2) \right] \|x_\star\|^4.$$

$\square$

## B  Proofs for the random spiked and generative models

We are now ready to prove our main results for random spiked models and generative networks with random weights. We begin by recalling the following fact on the WDC of a single Gaussian layer.

**Lemma 7** (Lemma 11 in [29]). *Fix $0 < \epsilon < 1$ and suppose $W \in \mathbb{R}^{n \times k}$ has i.i.d. $\mathcal{N}(0, 1/n)$ entries. Then if $n \geq C_\epsilon k \log k$, then with probability at least $1 - 8n \exp(-\gamma_\epsilon k)$, $W$ satisfies the WDC with constant $\epsilon$. Here $C_\epsilon$ and $\gamma_\epsilon^{-1}$ depend polynomially on $\epsilon^{-1}$.*

By a union bound over all layers, using the above result we can conclude that the WDC holds simultaneously for all layers of the network with probability at least $1 - \sum_{i=1}^d 8n_i e^{-\gamma_\epsilon n_{i-1}}$. Note in particular that this argument does not requires the independence of the layers.

By Lemma 7, with high probability the random generative network $G$ satisfies the WDC. Therefore if we can guarantee the assumptions on the noise term, then the proof of the main Theorem 2 follows from the deterministic Theorem 3 and the previous lemma.

Before turning to the bounds of the noise terms in the spiked models, we recall the following lemma which bounds the number of possible $\Lambda_x$ for $x \neq 0$. Note that this is related to the number of possible regions defined by a deep Relu network.

**Lemma 8** (Proof of Lemma 8 in [31]). *Consider a network $G$ as defined in (3) with $d \geq 2$, weight matrices $W_i \in \mathbb{R}^{n_i \times n_{i-1}}$ with i.i.d. entries $\mathcal{N}(0, 1/n_i)$ and $\log(10) \leq k/4 \log(n_1)$. Then, with probability one, for any $x \neq 0$ the number of different matrices $\Lambda_x$ is:*

$$|\{\Lambda_x | x \neq 0\}| \leq 10^{d^2} (n_1^d n_2^{d-1} \ldots n_d)^k \leq (n_1^d n_2^{d-1} \ldots n_d)^{2k}$$

In the next section we use this lemma to control the noise term $\Lambda_x^\intercal H \Lambda_x$ where:

- in the **Spiked Wishart Model** $H = \Sigma_N - \Sigma$;
- in the **Spiked Wigner Model** $H = \mathcal{H}$.

We then conclude in section B.3 with the proof of Proposition 1.B.

## B.1 Spike Wigner Model

Recall that in the Wigner model $Y = G(x_\star)G(x_\star)^\intercal + \mathcal{H}$ and the symmetric noise matrix $\mathcal{H}$ follows a *Gaussian Orthogonal Ensemble* $\text{GOE}(\nu, n)$, that is $\mathcal{H}_{ii} \sim \mathcal{N}(0, 2\nu/n)$ for all $1 \leq i \leq n$ and $\mathcal{H}_{ij} \sim \mathcal{N}(0, \nu/n)$ for $1 \leq j < i \leq n$. Our goal is to bound $\|\Lambda_x^\intercal \mathcal{H}\Lambda_x\|$ uniformly over $x$ with high probability.

Fix $x \in \mathbb{R}^k$, and let $\mathcal{N}_{1/4}$ be a $1/4$-net on the sphere $\mathcal{S}^{k-1}$ such that $|\mathcal{N}_{1/4}| \leq 9^k$ and:

$$\|\Lambda_x^\intercal \mathcal{H}\Lambda_x\| \leq 2 \max_{z \in \mathcal{N}_{1/4}} |\langle \Lambda_x^\intercal \mathcal{H}\Lambda_x z, z\rangle|.$$

For any $z \in \mathcal{N}_{1/4}$ let $\ell_{x,z} := \Lambda_x z \in \mathbb{R}^n$ and note that by the assumption on the entries of $\mathcal{H}$ it holds that $\ell_{x,z}^\intercal \mathcal{H}\ell_{x,z} \sim \mathcal{N}(0, \nu^2 \|\ell_{x,z}\|^4/n)$. In particular by Lemma 1, the quadratic form $\ell_{x,z}^\intercal \mathcal{H}\ell_{x,z}$ is sub-Gaussian with parameter $\gamma^2$ given by:

$$\gamma^2 := \frac{\nu^2}{n}\left(\frac{13}{12}\right)^2 \frac{1}{2^{2d}}.$$

Then for fixed $x \in \mathbb{R}^k$, standard sub-Gaussian tail bounds and a union bound over $\mathcal{N}_{1/4}$ give:

$$\mathbb{P}\big[\|\Lambda_x^\intercal \mathcal{H}\Lambda_x\| \geq 2u\big] \leq \mathbb{P}\big[\max_{z \in \mathcal{N}_{1/4}} \|\ell_{x,z}^\intercal \mathcal{H}\ell_{x,z}\| \geq u\big]$$

$$\leq \sum_{z \in \mathcal{N}_{1/4}} \mathbb{P}\big[\|\ell_{x,z}^\intercal \mathcal{H}\ell_{x,z}\| \geq u\big] \leq 2 \cdot 9^k e^{-\frac{u^2}{2\gamma^2}}.$$

Lemma 8, then ensures that the number of possible $\Lambda_x$ is at most $(n_1^d n_2^{d-1} \ldots n_d)^{2k}$, so a union bound over this set allows to conclude that:

$$\mathbb{P}\big[\|\Lambda_x^\intercal \mathcal{H}\Lambda_x\| \leq \frac{\nu}{2^d}\sqrt{\frac{30k\log(3\,n_1^d n_2^{d-1}\ldots n_d)}{n}}, \text{ for all } x\big] \geq 1 - 2e^{-k\log(n)}.$$

## B.2 Spike Wishart Model

Recall that the data $\{y_i\}_{i=1}^N$ are i.i.d. samples from $\mathcal{N}(0, \Sigma)$ where $\Sigma = G(x_\star)G(x_\star)^\intercal + \sigma^2 I_n$. In the minimization problem (4) we take $Y = \Sigma_N - \sigma^2 I_n$ where $\Sigma_N$ is the empirical covariance matrix. The symmetric noise matrix $H$ is then given by $H = \Sigma_N - \Sigma$ and by the Law of Large Numbers $H \to 0$ as $N \to \infty$. We bound $\|\Lambda_x^\intercal H\Lambda_x\|$ with high probability uniformly over $x \in \mathbb{R}^k$.

Fix $x \in \mathbb{R}^k$, let $\mathcal{N}_{1/4}$ be a $1/4$-net on the sphere $\mathcal{S}^{k-1}$ such that $|\mathcal{N}_{1/4}| \leq 9^k$, and notice that:

$$\|\Lambda_x^\intercal H\Lambda_x\| \leq 2 \max_{z \in \mathcal{N}_{1/4}} |z^\intercal \Lambda_x^\intercal H\Lambda_x z|.$$

By a union bound on $\mathcal{N}_{1/4}$ we obtain for any fixed $z \in \mathcal{N}_{1/4}$:

$$\mathbb{P}\big[\|\Lambda_x^\intercal H\Lambda_x\| \geq 2u\big] \leq 9^k \mathbb{P}\big[|z^T \Lambda_x^\intercal H\Lambda_x z| \geq u\big].$$

Let $\ell_x := \Lambda_x z$ and note that:

$$z^T \Lambda_x^\intercal H\Lambda_x z = \frac{1}{N}\sum_{i=1}^N (\ell_x^\intercal y_i)^2 - \mathbb{E}[(\ell_x^\intercal y_i)^2]$$

Since $s_i := \ell_x^\intercal y_i \sim \mathcal{N}(0, \gamma^2)$ where $\gamma^2 = \ell_x^\intercal \Sigma \ell_x$, then for $u/\gamma^2 \in (0, 1)$ by small deviation bounds for $\chi^2$ random variables (see for example [59, Example 2.11]):

$$\mathbb{P}\big[\|\Lambda_x^\intercal H\Lambda_x\| \geq 2u\big] \leq 9^k \mathbb{P}\big[|\frac{1}{N}\sum_{i=1}^N (s_i/\gamma)^2 - 1| \geq \frac{u}{\gamma^2}\big] \leq 2\exp\big[k\log 9 - \frac{N}{8}\frac{u^2}{\gamma^4}\big].$$

Recall now that $|\{\Lambda_x | x \neq 0\}| \leq (n_1^d n_2^d \ldots n_d)^k$, then proceeding as for the Wigner case by a union bound over all possible $\Lambda_x$:

$$\mathbb{P}\big[\|\Lambda_x^\intercal H\Lambda_x\| \leq 2\sqrt{\frac{24k\log(3\,n_1^d n_2^{d-1}\ldots n_d)}{N}}\gamma^2, \text{ for all } x\big] \geq 1 - 2e^{-k\log(3n)}$$

Similarly when $u/\gamma^2 \geq 1$ we obtain

$$\mathbb{P}\big[\|\Lambda_x^\intercal H\Lambda_x\| \leq 2\frac{24k\log(3\,n_1^d n_2^{d-1}\ldots n_d)}{N}\gamma^2, \text{ for all } x\big] \geq 1 - 2e^{-k\log(3n)}$$

### B.3 Proof of Proposition 1

Observe that Proposition 1.A follows from Proposition 2 after noticing that the assumptions on $\epsilon$ and $\omega$ in Theorem 2 imply that $\mathcal{B}(x_\star, r_+) \subset \mathcal{B}(x_\star, \varrho\|x_\star\|d^{-12})$ and $\mathcal{B}(-\rho_d x_\star, r_-) \subset \mathcal{B}(-\rho_d x_\star, \varrho\|x_\star\|d^{-12})$.

We next recall the following fact on the local Lipschitz property of the generative network.

**Lemma 9** (Lemma 21 in [31]). *Suppose* $x \in \mathcal{B}(x_\star, d\sqrt{\epsilon}\|x_\star\|)$, *and the WDC holds with* $\epsilon < 1/(200)^4/d^6$. *Then it holds that:*

$$\|G(x) - G(x_\star)\| \leq \frac{1.2}{2^{d/2}}\|x - x_\star\|.$$

The proof of Proposition 1.B follows now from $y_\star = G(x_\star)$, the above Lemma, the bounds (15) and (17) and the assumptions on $\epsilon$ and the noise term.