[Reviews · NeurIPS 2020]

Review 1

Summary and Contributions: The authors study the recovery problem for spiked Wigner/Wishart models. The spike is assumed to be rank-1 and generated by a d-layer feed forward network where the weights are Gaussian and the activation is ReLU. It is proved that the signal can be recovered with small mean-squared error, which is near-optimal, provided that the number of samples and the dimension of the signal are of same order (up to a log factor). In this case, the recovery can be done by simple algorithms such as gradient descent methods.

Strengths: The work considers the spike prior that is relevant to the neural network. The result is significant, in that it shows there is no computational-to-statistical gap in the rank-1 recovery problem under the assumption of the paper.

Weaknesses: In the analysis of spiked models, the signal-to-ratio (SNR) is the key parameter and sharp transitions may happen by varying it. In this work, the SNR is given by \sigma and \nu, and it affects the analysis (Theorem 2) via the noise variance \omega, which is not properly defined. It is thus hard to compare this work with prior works on the spiked models.

Correctness: The claims seem correct, but I did not check all the detail in the supplementary materials.

Clarity: The paper is clearly written in general, but there is room to be improved in some places (see below).

Relation to Prior Work: It is discussed well in Section 2.

Reproducibility: Yes

Additional Feedback: Many symbols are not properly defined or explained. Below are a few examples. \Sigma_N is not defined in line 75. It is implicitly assumed that n_d = n without any mention. \rho_d$ is not defined in Theorem 1. The noise variance \omega is unclear in Theorem 2. edit: After reading the other reviews and also the rebuttal, I agree that the lack of concrete algorithms regarding the "no computational-statistical gap" is an issue that should be resolved in the final version of the paper in some way. I still think that the result in the current manuscript is substantial and thus I did not changed my score after reading the rebuttal.


Review 2

Summary and Contributions: The paper considers the estimation of a rank-one matrix perturbed by Gaussian noise. Two models are studied: 1. the spiked Wishart model, where one observes Y = u y^T + \sigma Z, u and y being unknown vectors and Z a noise matrix 2. the spiked Wigner model, where Y = y y^T + \sigma Z. These problems have been extenseively in the case where y is sparse. In this regime, it is known that there exists a computational-to-statistical gap. The current paper consider the case where the signal y is produced by a (deep) ReLu neural network: y = G(x^*), where G is a neural network with ReLu activation functions. Recent works advocate for the absence of a computational gap in this setting. The present paper provide further evidences in that direction, showing that the landscape of the loss f(x) = ||G(x) G(x)^T - y||_F^2 does not contains other local minima than x=x^* or x = - r x^* for some r. Numerical simulations illustrate the results with perfect agreement.

Strengths: - The problem studied here is relevant: generative models are a very natural assumptions. - The authors manage to study the complicated loss function f and obtain sound results - While the main results are error bounds, the numerical simulations at the end of the paper suggest that these bounds capture the correct scaling of the problem.

Weaknesses: - The results are only about the landscape of the loss f(x), it would be nice to translate these results in terms of performance of algorithms. Ideally, it would be nice to have a result on the convergence time of gradient descent for minimizing f; instead of only the existence of descent directions. However, this might be much more involved.

Correctness: Yes

Clarity: Yes, the paper is clear and nicely written.

Relation to Prior Work: Yes

Reproducibility: Yes

Additional Feedback: Post-rebuttal edit: I think that this is a good submission: being able to characterize the landscape of a complicated cost function as the one studied in the paper is a significant achievement. The main weakness remains for me the absence of result about algorithms, and unfortunately, as one could expect and as the authors mentioned in their feedback, guarantees for gradient based methods seem to require a significant amount of work. He may be interesting to comment more precisely (than the authors already did in their feedback) about the ingredients missing to establish such results. I find the approach of the authors (proposing a gradient based algorithm and verifying that it performs well) pragmatic and convincing: they show strong evidences indicating the absence of computational gap.


Review 3

Summary and Contributions: This paper studies the nonasymptotic performance of rank-one matrix recovery, where the factors of the rank-one matrix is drawn from a generative model. The authors characterized the optimization landscape of a natural least-squares loss for the defined problem. They also proposed a subgradient algorithm which empirically achieves good performance. ===== Post rebuttal edit: thank you very much for the response. The paper will be much stronger with a polynomial time convergent algorithm to establish the claim on "breaking the computational-statistical barrier". While it is not strictly necessary to have an algorithm, nonetheless, the writing of this paper might need to be revised to make it clear where the contributions are. The authors are recommended to condense the literature review on sparse PCA and leave more room to discuss their own results.

Strengths: This paper follows [31] and [29] to study the landscape of a rank-one matrix recovery problem with generative models under the Wishart and Wigner models respectively. Some novel proof ingredients are needed, as suggested by the authors, in order to generalize the previous results to this new problem. However, the novel ingredients do not seem to be significant.

Weaknesses: - The paper is titled "low-rank" matrix recovery but it only deals with the rank-one case. A more faithful title would be appreciated. - It is unclear why this problem is comparable to the sparse PCA problem. It is understood that the computational barrier of sparse PCA problems is related to the planted clique conjecture, however I fail to see why the generative model setting a priori would suffer from a similar computational barrier (in other words, does the key barrier in sparse PCA also occurs in this problem, and somehow it can be broken to close the computation-statistical gap?) This aspect has not been clear to me, and the whole page the authors spent on the literature of sparse PCA becomes too long and somewhat irrelevant. - In order to claim "computation-statistical" guarantees, the authors need to prove that there exists an algorithm that solves the proposed problem in a polynomial time. Such an algorithm is not established in this paper. The subgradient algorithm is an empirical one. The discussions after Theorem 1 is not very sound, since having a descent direction does not mean a simple gradient-type algorithm can find it, and moreover, escape bad saddle points in polynomial time. Without a detailed study of the algorithm, "closing the computation-statistical gap" is an overclaim.

Correctness: The paper needs to show the existence of an efficient algorithm if they decide to claim both computational and statistical efficiencies. The theorem does not specify the size of the gradient (I wasn't able to distill it clearly from the proofs). In fact, due to the nonsmooth aspect of the loss function, I don't think existing saddle-point escaping algorithms can be applied directly. Therefore, it is premature for the authors to claim the computational efficiency of the approach.

Clarity: The paper is clear for most parts, though the proof sketch is notation heavy and hard-to-follow. Some notation appear undefined, such as D_{-v}f.

Relation to Prior Work: The discussions to prior work are adequate.

Reproducibility: Yes

Additional Feedback:


Review 4

Summary and Contributions: The paper focuses on low-rank matrix recovery based on parametrizations using trained generative neural networks. The authors prove that this parametrization -- contrary to sparse parametrizations -- yields optimal error rates which can be obtained by tractable gradient descent type of algorithms. The main contributions are to provide such an analysis, showing that such generative priors have no computational-to-statistical gap in the low-rank matrix recovery problem.

Strengths: - The problem under study, statistical parametrization using neural netwrks, is important as well as its application to low-rank recovery. The proposed approach is novel and original, and paves the way to new developments in low-rank matrix recovery. - The theoretical results are sound and original : to my knowledge the authors are the first to provide such theoretical analysis of low-rank recovery using GNN priors. - The empirical results look promising and open possibilities of application However, there are several limitations as detailed below

Weaknesses: - I am not sure whether the proof techniques are novel as the authors mention that they use arguments of [29]: could they comment on this point ? - The impact of the GNN parametrization is not completely clear : there is no computational-statistical gap, but the factor d^10 seems huge, and leads to rates much larger than those obtained by sparse parametrization, even with polynomial time algorithms. Could the authors comment on that ? - The empirical study is very limited, with no comparison to other methods (the authors just display their own results), and no evaluation on applied problems. - There are many typos and odd formulations in the article, and it should be carefully read to correct them. E.g. l.81 math inserted in middle of sentence ; l. 153 "as recovery a spike from a spiked random matrix models"; l.154 "simultaneously both"; l. 195 "Assume Assumption 1 ", etc. - The theoretical results are hard to read because of the way theorems and propositions are written, with notation introduced inside. I recommend to simplify them (put notation and assumptions outside), and to comment the results much more : there is almost no discussion of the meaning of the results and comparison to existing work after TH. 2 and Prop.1

Correctness: The claims and theoretical results are sound, and the empirical methodology correct

Clarity: The paper is well written, but there are many typos, and the theretical statements are hard to follow.

Relation to Prior Work: The discussion with previous contributions is clear

Reproducibility: Yes

Additional Feedback:


Review 5

Summary and Contributions: This paper studies reconstruction of a low-rank component when it lies in the range of a generative model. This can be seen as an extension of Sparse PCA where sparsity is replaced with a vector in the range of a deep generative model a-la Bora et al. The mathematical technology is extending Hand and Voroninski [29] in a natural but non-trivial way.

Strengths: The problem is important and extends classical sparsity-based work for deep-learning models. The theoretical results are very interesting and technically deep.

Weaknesses: The setup is a bit artificial and I do not see that many situations where the low-rank component would have complex structure but this may not turn out to be the case. The empirical evaluation is limited, also because it is not that easy to find good problems with the proposed structure, I suppose.

Correctness: Results seem correct.

Clarity: The paper is reasonably well written but with several typos.

Relation to Prior Work: yes

Reproducibility: Yes

Additional Feedback: This is a solid theoretical contribution in a very interesting area. I would love a better empirical evaluation and a convincing application domain, but as a theory paper this is very good.

[Author Response · NeurIPS 2020]

– – – **Response to Reviewer 1** – – –

**Re: Definition of** $\omega$ The noise level $\omega$ is defined in Theorem 2 and it is analogous to the effective SNR $\sigma^2\sqrt{s\log(n)/N}$ which governs sharp transitions in Sparse PCA (e.g. Amini & Wainwright, 2009). In our case the effective SNR $\omega$ naturally depends also the inner layer dimensions and depth of the generative network. Up to log factors and polynomials in $d$, $\omega$ takes the form $\sigma^2\sqrt{k\log(n)/N}$ (see for example the informal Theorem 1). We will add a remark after the main Theorem 2 and Proposition 1 explaining how to interpret $\omega$ as an SNR making a connection with the bounds in previous works (in Section 2).

**Re: Symbols not properly defined** We regret the omission of the definitions of $n_d$ and other symbols. We will make sure they are properly defined and fix the typos in the camera ready.

– – – **Response to Reviewer 2** – – –

**Re: Proof of convergence** We suspect our analysis could indeed be translated into a convergence proof. Doing so would require significant technical enhancements, including establishing convexity (or a similar property) locally around the global minimizer, along with careful estimates to ensure a proper step size. These are significant technical improvements, and we leave them for future work.

– – – **Response to Reviewer 3** – – –

**Re: Novelty of the proof ingredients** The extension from previous works is substantial at a technical level. Both [29] and [31] solve quadratic objectives, but in our paper, the objective is quartic. Generally, quartic objectives are challenging to analyze because the fourth power of Gaussians have tails too thick for uniform concentration results. Our work demonstrates how to deal with these terms while maintaining optimal sample complexity. Additionally, unlike in [29] and [31], we control sub-exponential matrix noise with multiple structures. We will add a paragraph to the paper discussing exactly these points.

**Re: Title** We will change the title to "Nonasymptotic Guarantees for Spiked Matrix Recovery with Generative Priors"

**Re: Comparability to Sparse PCA** Signal recovery problems where multiple signal structures hold simultaneously (e.g. low-rank AND sparse matrices) have been notoriously difficult, leading to no tractable algorithms at optimal sample complexity. Consequently, one would expect that enforcing low-rank AND generative priors would be comparably difficult. In this work, we indeed show that this combination of structural priors is not inherently difficult. This would lead practitioners to invest in building and using generative priors, as studied in this paper.

**Re: Claiming computational-statistical guarantees** The central property that allows the "no computational-statistical gap" statement is the fact that the optimization landscape is benign. Specifically, we show that the direction given by the gradient (almost everywhere) is a descent direction (with nonzero directional derivative). It is true that we do not provide a proof of convergence of a particular algorithm, but we establish the the conditions are appropriate for such an algorithm to converge. See response to Reviewer 2 for further comments on a possible convergence proof.

**Re: gradient algorithm might not find descent direction** The theorem asserts that there is a linear descent direction (with rate bounded away from zero) for any direction within the Clarke subdifferential (which almost everywhere is precisely the gradient). Thus, the descent claim applies in any direction a subgradient descent algorithm would take. Additionlly, this result ensures that there are no spurious maxima or saddles that can not be escaped.

– – – **Response to Reviewer 4** – – –

**Re: Novelty of the proof ingredients** See response to Reviewer 3.

**Re: Polynomial scaling in** $d$ **of the rates** We focused on studying optimal sample complexity with respect to the intrinsic dimensionality of the signal. We aimed for a result where the dependence on depth $d$ is polynomial (it could have been exponential: for example straightforward bounds on the Lipschitness of a network are exponential in the parameter $d$!) Optimizing the proof for superior dependence on $d$ would not drastically alter the fundamental theoretical advance, and it would require a much more cumbersome proof. As we show in the numerical experiments the bounds are quite conservative and the actual dependence on the depth is much better in practice.

**Re: Empirical results** This paper is a theoretical paper of a fundamental sample complexity improvement when using generative priors. We agree it would be exciting to see demonstrations in practical domain-specific applications, and we hope this result inspires authors to go through the difficult process of training models in these settings.

**Re: Clarity, typos and odd formulations** We will simplify the presentation of the main results by putting notation and assumptions outside. We will add a comment after the main Theorem 2 and Proposition 1 explaining their consequences and interpretation.

[Meta-Review · NeurIPS 2020]

The reviewers appreciate the analysis showing that the landscape of the optimization problem has tractable structure despite the complexity of the model. This is similar to the Hand-Voroninski result (extended to SPCA) but still remains one of the most impressive theoretical phenomena in optimization for deep networks. Reviewers and meta-reviewer were concerned that this does not lead (yet) to a proof that gradient-based optimization will converge to the global opt in poly-time but we expect this result to also be obtained. The empirical evaluation is limited and the problem is a bit contrived but perhaps the authors or someone else in the community can find a good application for this type of structure. There was a debate if the authors oversell their lack of "statistical to computational gap" but the meta-reviewer thinks that this is established in an average-case sense. For this reason this is a great fit for Neurips.